# Bigger is not Always Better:
# Scaling Properties of Latent Diffusion Models

**Kangfu Mei**[*]                                    *kmei1@jhu.edu*
*Johns Hopkins University*

**Zhengzhong Tu**[†]                                  *tzz@tamu.edu*
*Texas A&M University*

**Mauricio Delbracio**                               *mdelbra@google.com*
*Google*

**Hossein Talebi**                                   *htalebi@google.com*
*Google*

**Vishal M. Patel**                                  *vpatel36@jhu.edu*
*Johns Hopkins University*

**Peyman Milanfar**                                  *milanfar@google.com*
*Google*

**Reviewed on OpenReview:** *https://openreview.net/forum?id=0u7pWfjri5*

## Abstract

We study the scaling properties of latent diffusion models (LDMs) with an emphasis on their sampling efficiency. While improved network architecture and inference algorithms have shown to effectively boost sampling efficiency of diffusion models, the role of model size—a critical determinant of sampling efficiency—has not been thoroughly examined. Through empirical analysis of established text-to-image diffusion models, we conduct an in-depth investigation into how model size influences sampling efficiency across varying sampling steps. Our findings unveil a surprising trend: when operating under a given inference budget, smaller models frequently outperform their larger equivalents in generating high-quality results. Moreover, we extend our study to demonstrate the generalizability of the these findings by applying various diffusion samplers, exploring diverse downstream tasks, evaluating post-distilled models, as well as comparing performance relative to training compute. These findings open up new pathways for the development of LDM scaling strategies which can be employed to enhance generative capabilities within limited inference budgets.

## 1 Introduction

Latent diffusion models (LDMs) (Rombach et al., 2022), and diffusion models in general, trained on large-scale, high-quality data (Lin et al., 2014; Schuhmann et al., 2022) have emerged as a powerful and robust framework for generating impressive results in a variety of tasks, including image synthesis and editing (Rombach et al., 2022; Podell et al., 2023; Delbracio & Milanfar, 2023; Ren et al., 2023; Qi et al., 2023), video creation (Mei & Patel, 2023; Mei et al., 2023; Wu et al., 2023; Singer et al., 2022), audio production (Liu et al., 2023a), and 3D synthesis (Lin et al., 2023; Liu et al., 2023b). Despite their versatility, the major

---

[*]This work was done during an internship at Google.
[†]This work was done while at Google.

barrier against wide deployment in real-world applications (Du et al., 2023; Choi et al., 2023) comes from their low *sampling efficiency*. The essence of this challenge lies in the inherent reliance of LDMs on multi-step sampling (Song et al., 2021b; Ho et al., 2020) to produce high-quality outputs, where the total cost of sampling is the product of sampling steps and the cost of each step. Specifically, the go-to approach involves using the 50-step DDIM sampling (Song et al., 2021a; Rombach et al., 2022), a process that, despite ensuring output quality, still requires a relatively long latency for completion on modern mobile devices with post-quantization. In contrast to single shot generative models (e.g., generative-adversarial networks (GANs) (Goodfellow et al., 2020)) which bypass the need for iterative refinement (Goodfellow et al., 2020; Karras et al., 2019), the operational latency of LDMs calls for a pressing need for efficiency optimization to further facilitate their practical applications.

Recent advancements in this field (Li et al., 2023; Zhao et al., 2023; Peebles & Xie, 2023; Kim et al., 2023b;a; Choi et al., 2023) have primarily focused on developing faster network architectures with comparable model size to reduce the inference time per step, along with innovations in improving sampling algorithms that allow for using less sampling steps (Song et al., 2021a; Dockhorn et al., 2022; Karras et al., 2022; Lu et al., 2022a; Liu et al., 2023c; Xu et al., 2023). Further progress has been made through diffusion-distillation techniques (Luhman & Luhman, 2021; Salimans & Ho, 2022; Song et al., 2023; Sauer et al., 2023b; Gu et al., 2023; Mei et al., 2024a), which simplifies the process by learning multi-step sampling results in a single forward pass, and then broadcasts this single-step prediction multiple times. These distillation techniques leverage the redundant learning capability in LDMs, enabling the distilled models to assimilate additional distillation knowledge. Despite these efforts being made to improve diffusion models, the sampling efficiency of smaller, less redundant models has not received adequate attention. A significant barrier to this area of research is the scarcity of available modern accelerator clusters (Jouppi et al., 2023), as training high-quality text-to-image (T2I) LDMs from scratch is both time-consuming and expensive—often requiring several weeks and hundreds of thousands of dollars.

In this paper, we empirically investigate the scaling properties of LDMs, with a particular focus on understanding how their scaling properties impact the sampling efficiency across various model sizes. We trained a suite of 12 text-to-image LDMs from scratch, ranging from 39 million to 5 billion parameters, under a constrained budget. Example results are depicted in Fig. 1. All models were trained on TPUv5 using internal data sources with about 600 million aesthetically-filtered text-to-image pairs. Our study reveals that there exist a scaling trend within LDMs, notably that smaller models may have the capability to surpass larger models under an equivalent sampling budget. Furthermore, we investigate how the size of pre-trained text-to-image LDMs affects their sampling efficiency across diverse downstream tasks, such as real-world super-resolution (Saharia et al., 2022; Sahak et al., 2023) and subject-driven text-to-image synthesis (i.e., Dreambooth) (Ruiz et al., 2023).

## 1.1 Summary

Our key findings for scaling latent diffusion models in text-to-image generation and various downstream tasks are as follows:

**Pretraining performance scales with training compute.** We demonstrate a clear link between compute resources and LDM performance by scaling models from 39 million to 5 billion parameters. This suggests potential for further improvement with increased scaling. See Section 3.1 for details.

**Downstream performance scales with pretraining.** We demonstrate a strong correlation between pretraining performance and success in downstream tasks. Smaller models, even with extra training, cannot fully bridge the gap created by the pretraining quality of larger models. This is explored in detail in Section 3.2.

**Smaller models sample more efficient.** Smaller models initially outperform larger models in image quality for a given sampling budget, but larger models surpass them in detail generation when computational constraints are relaxed. This is further elaborated in Section 3.3.1 and Section 3.3.2.

**Sampler does not change the scaling efficiency.** Smaller models consistently demonstrate superior sampling efficiency, regardless of the diffusion sampler used. This holds true for deterministic DDIM (Song et al., 2021a), stochastic DDPM (Ho et al., 2020), and higher-order DPM-Solver++ (Lu et al., 2022b). For more details, see Section 3.4.

**Smaller models sample more efficient on the downstream tasks with fewer steps.** The advantage of smaller models in terms of sampling efficiency extends to the downstream tasks when using less than 20 sampling steps. This is further elaborated in Section 3.5.

**Diffusion distillation does not change scaling trends.** Even with diffusion distillation, smaller models maintain competitive performance against larger distilled models when sampling budgets are constrained. This suggests distillation does not fundamentally alter scaling trends. See Section 3.6 for in-depth analysis.

## 2   Related Work

**Scaling laws.**   Recent Large Language Models (LLMs) including GPT (Brown et al., 2020), PaLM (Anil et al., 2023), and LLaMa (Touvron et al., 2023) have dominated language generative modeling tasks. The foundational works (Kaplan et al., 2020; Brown et al., 2020; Hoffmann et al., 2022) for investigating their scaling behavior have shown the capability of predicting the performance from the model size. They also investigated the factors that affect the scaling properties of language models, including training compute, dataset size and quality, learning rate schedule, etc. Those experimental clues have effectively guided the later language model development, which have led to the emergence of several parameter-efficient LLMs (Hoffmann et al., 2022; Touvron et al., 2023; Zhou et al., 2023; Alabdulmohsin et al., 2024). However, scaling generative text-to-image models are relatively unexplored, and existing efforts have only investigated the scaling properties on small datasets or small models, like scaling UNet (Nichol & Dhariwal, 2021) to 270 million parameters and DiT (Peebles & Xie, 2023) on ImageNet (14 million), or less-efficient autoregressive models (Chen et al., 2020). Different from these attempts, our work investigates the scaling properties by scaling down the efficient and capable diffusion models, *i.e.* LDMs (Rombach et al., 2022), on internal data sources that have about 600 million aesthetics-filtered text-to-image pairs for featuring the sampling efficiency of scaled LDMs. We also scale LDMs on various scenarios such as finetuning LDMs on downstream tasks (Wang et al., 2021; Ruiz et al., 2023) and distilling LDMs (Mei et al., 2024a) for faster sampling to demonstrate the generalizability of the scaled sampling-efficiency.

**Efficient diffusion models.**   Nichol et al. (Nichol & Dhariwal, 2021) show that the generative performance of diffusion models improves as the model size increases. Based on this preliminary observation, the model size of widely used LDMs, *e.g.*, Stable Diffusion (Rombach et al., 2022), has been empirically increased to billions of parameters (Ramesh et al., 2022; Podell et al., 2023). However, such a large model makes it impossible to fit into the common inference budget of practical scenarios. Recent work on improving the sampling efficiency focus on improving network architectures (Li et al., 2023; Zhao et al., 2023; Peebles & Xie, 2023; Kim et al., 2023b;a; Choi et al., 2023; Mei et al., 2024b) or the sampling procedures (Song et al., 2021a; Dockhorn et al., 2022; Karras et al., 2022; Lu et al., 2022a; Liu et al., 2023c; Xu et al., 2023; Mei et al., 2025). We explore sampling efficiency by training smaller, more compact LDMs. Our analysis involves scaling down the model size, training from scratch, and comparing performance at equivalent inference cost.

**Efficient non-diffusion generative models.**   Compared to diffusion models, other generative models such as, Variational Autoencoders (VAEs) (Kingma & Welling, 2014; Rezende & Mohamed, 2015; Makhzani et al., 2015; Vahdat & Kautz, 2020), Generative Adversarial Networks (GANs) (Goodfellow et al., 2020; Mao et al., 2017; Karras et al., 2019; Reed et al., 2016; Miyato et al., 2018), and Masked Models (Devlin et al., 2019; Raffel et al., 2020; He et al., 2022; Chang et al., 2022; 2023), are more efficient, as they rely less on an iterative refinement process. Sauer et al. (Sauer et al., 2023a) recently scaled up StyleGAN (Karras et al., 2019) into 1 billion parameters and demonstrated the single-step GANs' effectiveness in modeling text-to-image generation. Chang et al. (Chang et al., 2023) scaled up masked transformer models for text-to-image generation. These non-diffusion generative models can generate high-quality images with less inference cost,

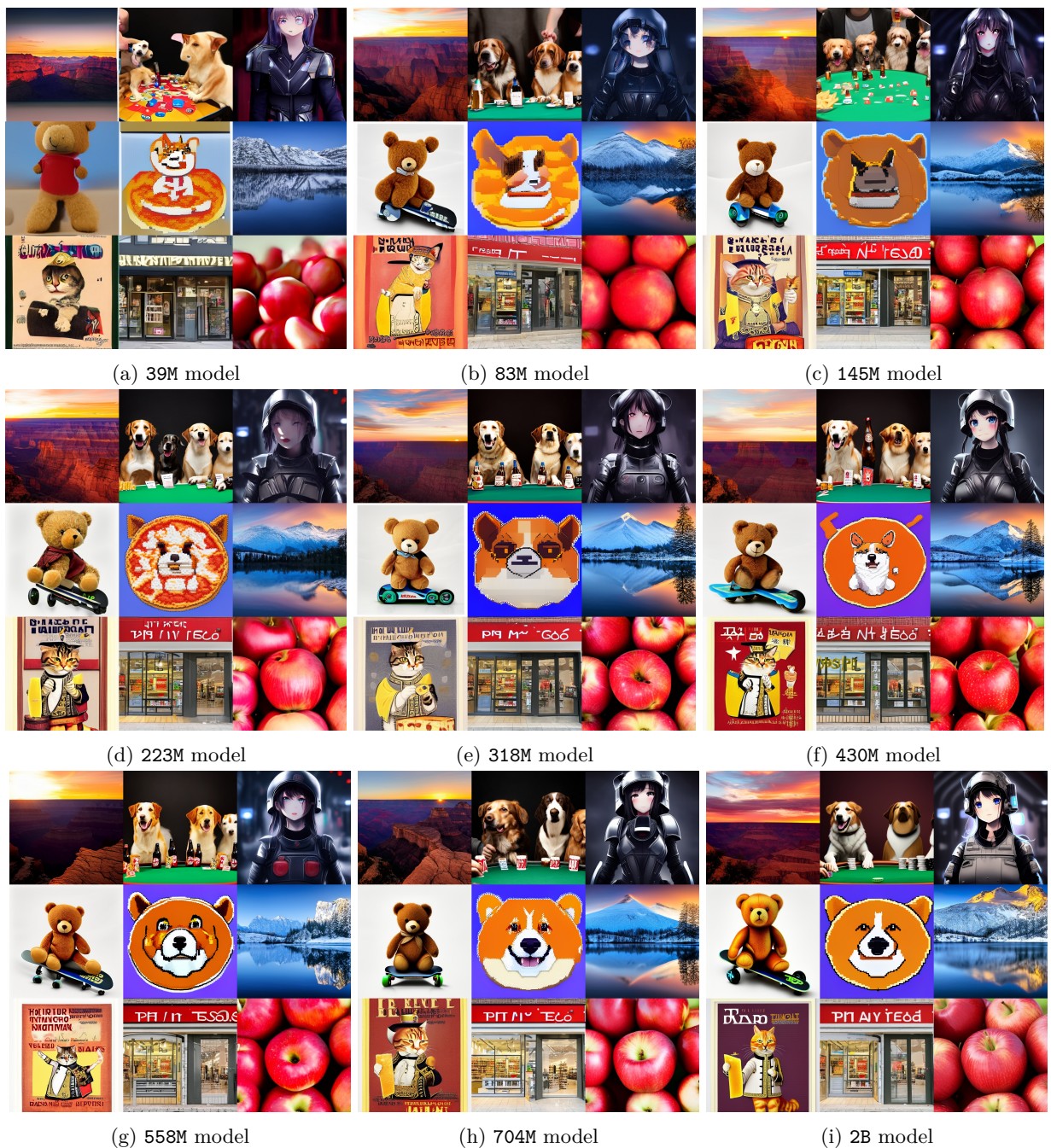

(a) 39M model        (b) 83M model        (c) 145M model

(d) 223M model        (e) 318M model        (f) 430M model

(g) 558M model        (h) 704M model        (i) 2B model

Figure 1: Text-to-image results from our scaled LDMs (39M - 2B), highlighting the improvement in visual quality with increased model size (note: 39M model is the exception). All images generated using 50-step DDIM sampling and CFG rate of 7.5. We use representative prompts from PartiPrompts Yu et al. (2022), including *"a professional photo of a sunset behind the grand canyon."*, *"Dogs sitting around a poker table with beer bottles and chips. Their hands are holding cards."*, *'Portrait of anime girl in mechanic armor in night Tokyo."*, *"a teddy bear on a skateboard."*, *"a pixel art corgi pizza."*, *"Snow mountain and tree reflection in the lake."*, *"a propaganda poster depicting a cat dressed as french emperor napoleon holding a piece of cheese."*, *"a store front that has the word 'LDMs' written on it."*, and *"ten red apples."*. Check our supplement for additional visual comparisons.

| Params | 39M | 83M | 145M | 223M | 318M | 430M | 558M | 704M | 866M | 2B | 5B |
|---|---|---|---|---|---|---|---|---|---|---|---|
| Filters ($c$) | 64 | 96 | 128 | 160 | 192 | 224 | 256 | 288 | 320 | 512 | 768 |
| GFLOPS | 25.3 | 102.7 | 161.5 | 233.5 | 318.5 | 416.6 | 527.8 | 652.0 | 789.3 | 1887.5 | 4082.6 |
| Norm. Cost | 0.07 | 0.13 | 0.20 | 0.30 | 0.40 | 0.53 | 0.67 | 0.83 | 1.00 | 2.39 | 5.17 |
| FID ↓ | 25.30 | 24.30 | 24.18 | 23.76 | 22.83 | 22.35 | 22.15 | 21.82 | 21.55 | 20.98 | 20.14 |
| CLIP ↑ | 0.305 | 0.308 | 0.310 | 0.310 | 0.311 | 0.312 | 0.312 | 0.312 | 0.312 | 0.312 | 0.314 |

Table 1: We scale the baseline LDM (*i.e.*, 866M Stable Diffusion v1.5) by changing the base number of channels $c$ that controls the rest of the U-Net architecture as $[c, 2c, 4c, 4c]$ (See Fig. 2). GFLOPS are measured for an input latent of shape $64 \times 64 \times 4$ with FP32. We also show a normalized running cost with respect to the baseline model. The text-to-image performance (FID and CLIP scores) for all scaled LDMs is evaluated on the COCO-2014 validation set with 30k samples, using 50-step DDIM sampling and Classifier-free Guidance (CFG) with a rate of 7.5. It is worth noting that all the model sizes, and the training and the inference costs reported in this work only refer to the denoising UNet in the latent space, and do not include the 1.4B text encoder and the 250M latent encoder and decoder.

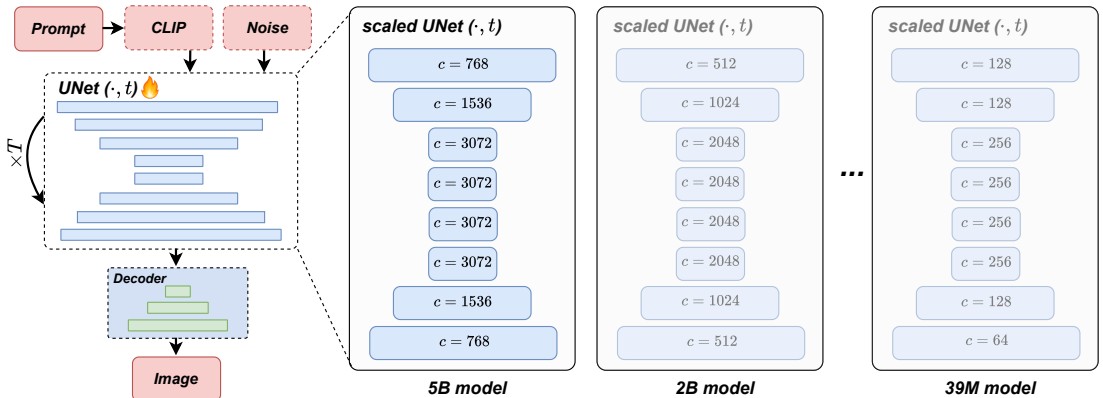

Figure 2: Our scaled latent diffusion models vary in the number of filters within the denoising U-Net. Other modules remain consistent. Smooth channel scaling (64 to 768) within residual blocks yields models ranging from 39M to 5B parameters. For downstream tasks requiring image input, we use an encoder to generate a latent code; this code is then concatenated with the noise vector in the denoising U-Net.

which require fewer sampling steps than diffusion models and autoregressive models, but they need more parameters, *i.e.*, 4 billion parameters.

## 3    Scaling LDMs

We developed a family of powerful Latent Diffusion Models (LDMs) built upon the widely-used 866M Stable Diffusion v1.5 standard (Rombach et al., 2022)[1]. The denoising UNet of our models offers a flexible range of sizes, with parameters spanning from 39M to 5B. We incrementally increase the number of filters in the residual blocks while maintaining other architecture elements the same, enabling a predictably controlled scaling. Table 1 shows the architectural differences among our scaled models. We also provide the relative cost of each model against the baseline model. Fig. 2 shows the architectural differences during scaling. Models were trained using the web-scale aesthetically filtered text-to-image dataset, *i.e.*, WebLI (Chen et al., 2022). All the models are trained for 500K steps, batch size 2048, and learning rate 1e-4. This allows for all the models to have reached a point where we observe diminishing returns. Fig. 1 demonstrates the consistent generation capabilities across our scaled models. We used the common practice of 50 sampling

---

[1]We adopted SD v1.5 since it is among the most popular diffusion models https://huggingface.co/models?sort=likes.

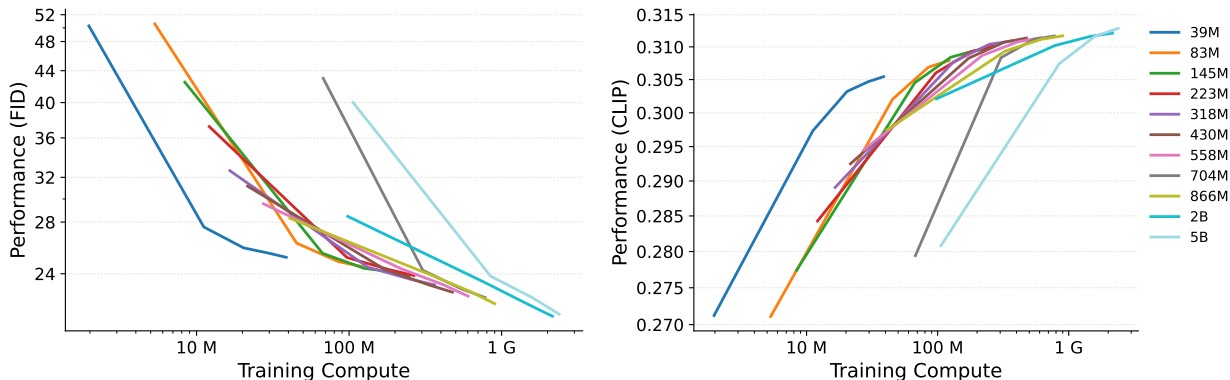

Figure 3: In text-to-image generation using 50-step DDIM sampling and CFG rate of 7.5, we observe consistent trends across various model sizes in how quality metrics (FID and CLIP scores) relate to training compute (*i.e.*, the total GFLOPS spend on training). Under moderate training resources, training compute is the most relevant factor dominating quality.

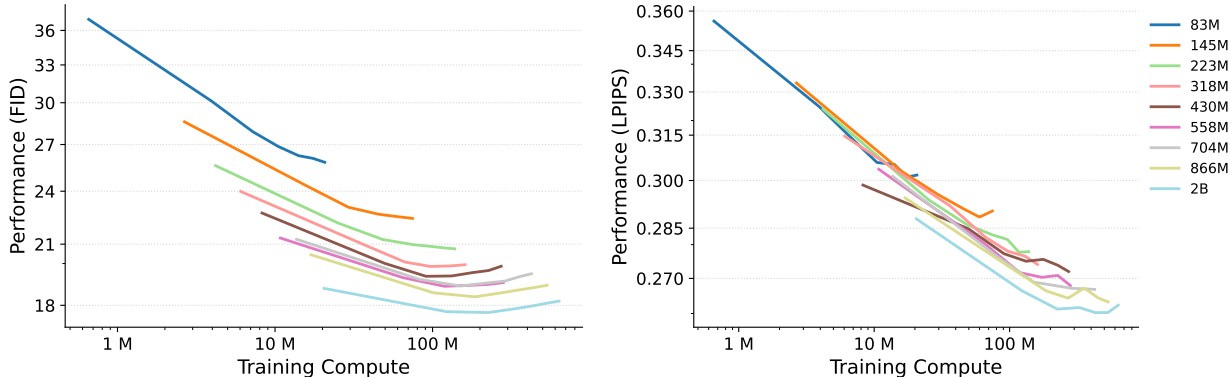

Figure 4: In 4× real image super-resolution using 50-step DDIM sampling, FID and LPIPS scores reveal an interesting divergence. Model size drives FID score improvement, while training compute most impacts LPIPS score. Despite this, visual assessment (Fig. 5) confirms the importance of model size for superior detail recovery (similarly as observed in the text-to-image pretraining).

steps with the DDIM sampler, 7.5 classifier-free guidance rate, for text-to-image generation. The visual quality of the results exhibits a clear improvement as model size increases.

In order to evaluate the performance of the scaled models, we test the text-to-image performance of scaled models on the validation set of COCO 2014 (Lin et al., 2014) with 30k samples. For downstream performance, specifically real-world super-resolution, we test the performance of scaled models on the validation of DIV2K with 3k randomly cropped patches, which are degraded with the RealESRGAN degradation (Wang et al., 2021).

## 3.1 Training compute scales text-to-image performance

We find that our scaled LDMs, across various model sizes, exhibit similar trends in generative performance relative to training compute cost, especially after training stabilizes, which typically occurs after 200K iterations. These trends demonstrate a smooth scaling in learning capability between different model sizes. To elaborate, Fig. 3 illustrates a series of training runs with models varying in size from 39 million to 5 billion parameters, where the training compute cost is quantified as the product of relative cost shown in Table 1 and training iterations. Model performance is evaluated by using the same sampling steps and sampling parameters. In scenarios with moderate training compute (i.e., $< 1G$, see Fig. 3), the generative performance of T2I models scales well with additional compute resources.

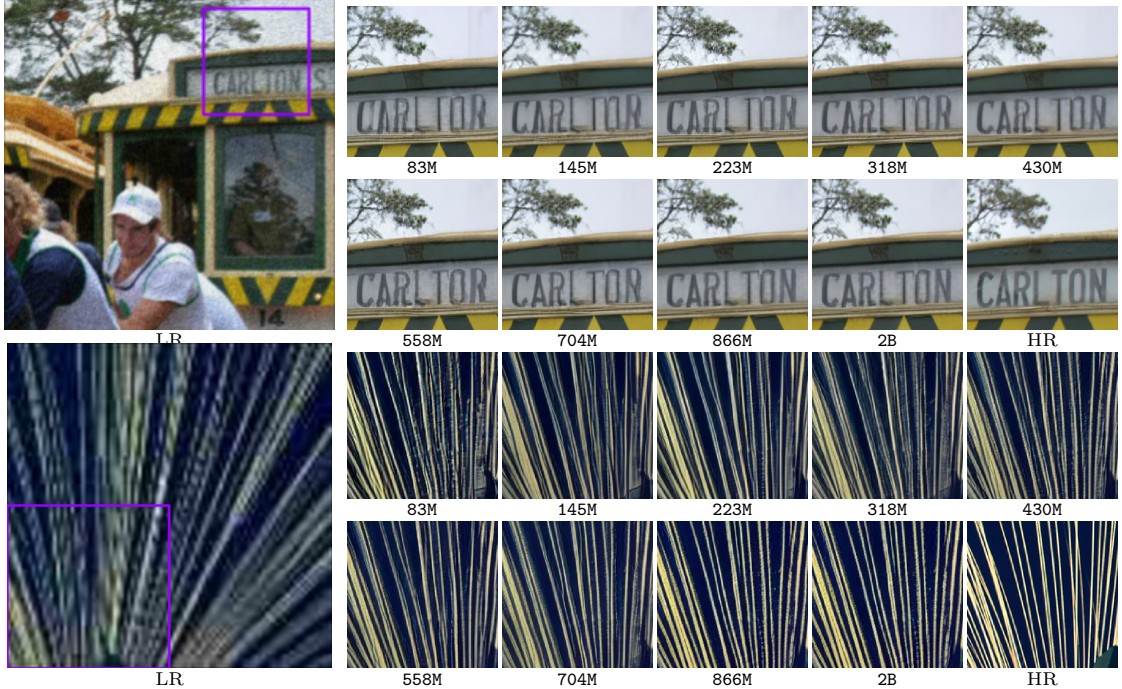

Figure 5: In 4× super-resolution using 50-step DDIM sampling, visual quality directly improves with increased model size. As these scaled models vary in pretraining performance, the results clearly demonstrate that pretraining boosts super-resolution capabilities in both quantitative (Fig 4) and qualitative ways. *Additional results are given in supplementary material.*

## 3.2 Pretraining scales downstream performance

Using scaled models based on their pretraining on text-to-image data, we finetune these models on the downstream tasks of real-world super-resolution (Saharia et al., 2022; Sahak et al., 2023) and DreamBooth (Ruiz et al., 2023). The performance of these pretrained models is shown in Table. 1. In the left panel of Fig. 4, we present the generative performance FID versus training compute on the super-resolution (SR) task. It can be seen that the performance of SR models is more dependent on the model size than training compute. Our results demonstrate a clear limitation of smaller models: they cannot reach the same performance levels as larger models, regardless of training compute.

While the distortion metric LPIPS shows some inconsistencies compared to the generative metric FID (Fig. 4), Fig. 5 clearly demonstrates that larger models excel in recovering fine-grained details compared to smaller models.

The key takeaway from Fig. 4 is that large super-resolution models achieve superior results even after short finetuning periods compared to smaller models. This suggests that pretraining performance (dominated by the pretraining model sizes) has a greater influence on the super-resolution FID scores than the duration of finetuning (*i.e.*, training compute for finetuning).

Furthermore, we compare the visual results of the DreamBooth finetuning on the different models in Fig. 6. We observe a similar trend between visual quality and model size. *Please see our supplement for more discussions on the other quality metrics.*

## 3.3 Scaling sampling-efficiency

### 3.3.1 Analyzing the effect of CFG rate.

Text-to-image generative models require nuanced evaluation beyond single metrics. Sampling parameters are vital for customization, with the Classifier-Free Guidance (CFG) rate (Ho & Salimans, 2022) directly

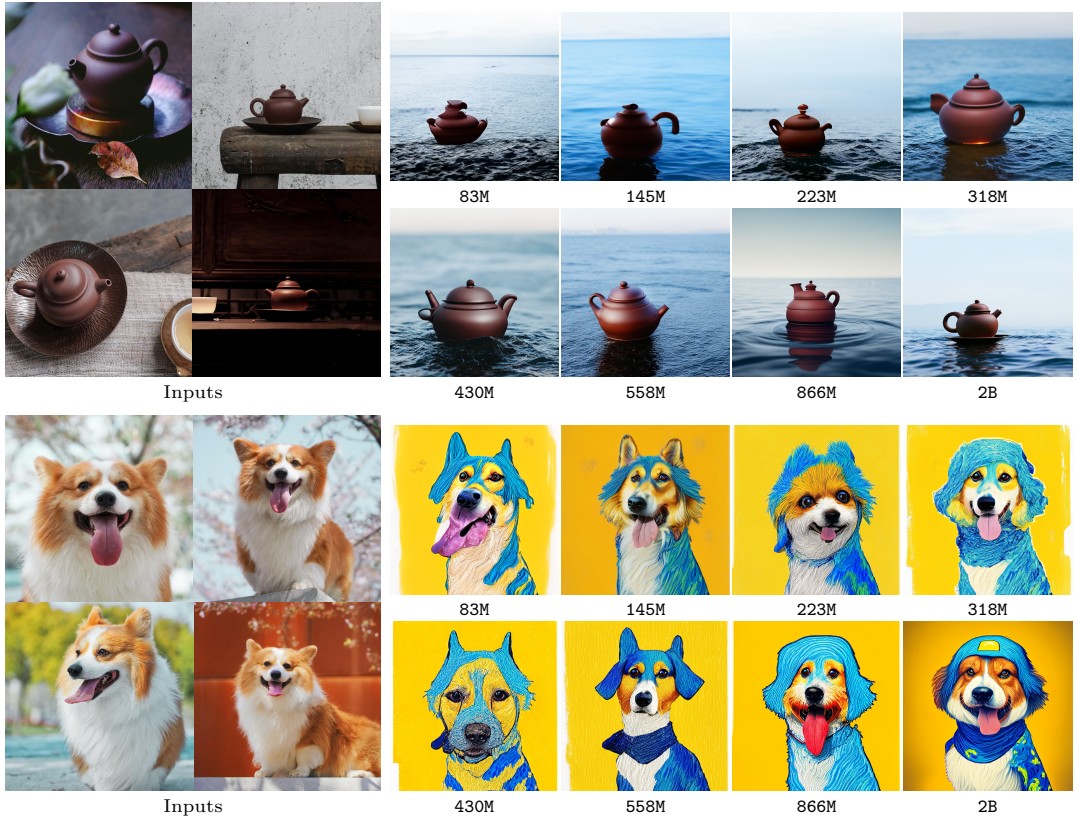

83M          145M          223M          318M

Inputs       430M          558M          866M          2B

Figure 6: Visualization of the Dreambooth results (using 50-step DDIM sampling and CFG rate of 7.5) shows two distinct tiers based on model size. Smaller models (83M-223M) perform similarly, as do larger ones (318M-2B), with a clear quality advantage for the larger group. *Additional results are given in supplementary material.*

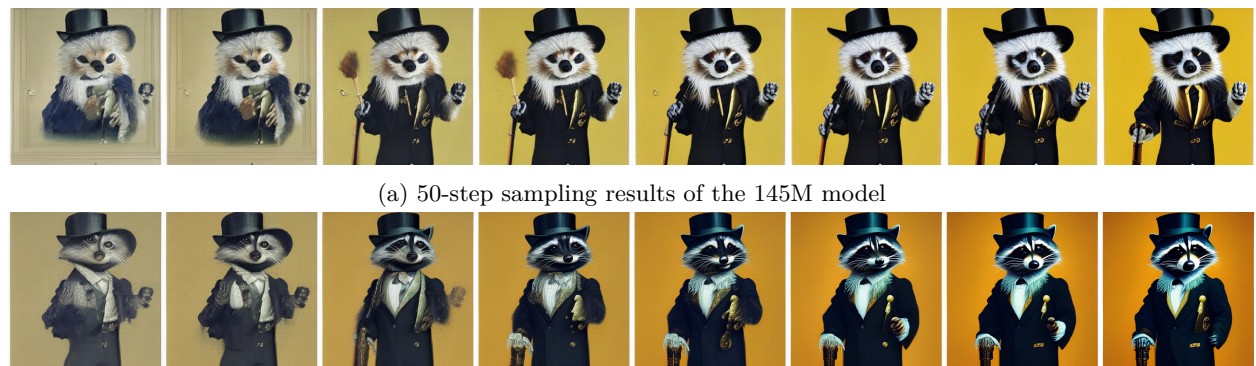

(a) 50-step sampling results of the 145M model

(b) 50-step sampling results of the 866M model

Figure 7: Visualization of text-to-image results with 50-step DDIM sampling and different CFG rates (from left to right in each row: $(1.5, 2.0, 3.0, 4.0, 5.0, 6.0, 7.0, 8.0)$). The prompt used is "*A raccoon wearing formal clothes, wearing a top hat and holding a cane. Oil painting in the style of Rembrandt.*". We observe that changes in CFG rates impact visual quality more significantly than the prompt semantic accuracy. We use the FID score for quantitative determination of optimal sampling performance (Fig. 8) because it directly measures visual quality, unlike the CLIP score, which focuses on semantic similarity.

influencing the balance between visual fidelity and semantic alignment with text prompt. Rombach et al. (Rombach et al., 2022) experimentally demonstrate that different CFG rates result in different CLIP and FID scores.

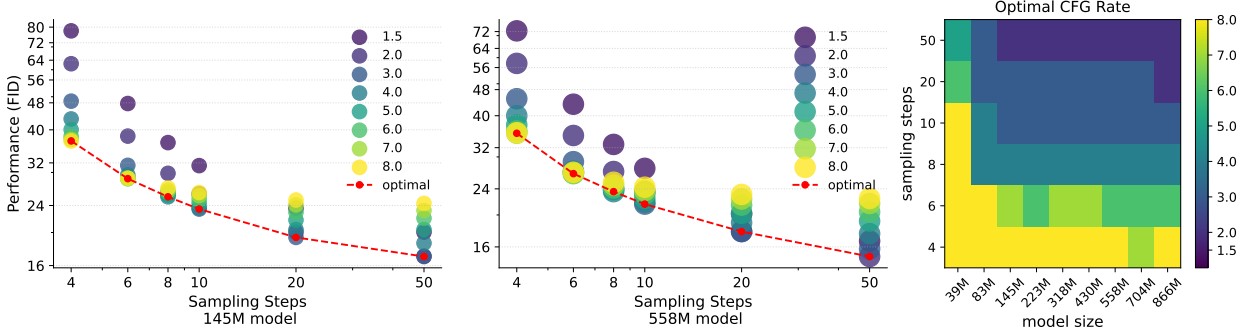

Figure 8: The impact of the CFG rate on text-to-image generation depends on the model size and sampling steps. As demonstrated in the left and center panels, the optimal CFG rate changes as the sampling steps increased. To determine the optimal performance (according to the FID score) of each model and each sampling steps, we systematically sample the model at various CFG rates and identify the best one. As a reference of the optimal performance, the right panel shows the CFG rate corresponding to the optimal performance of each model for a given number of sampling steps.

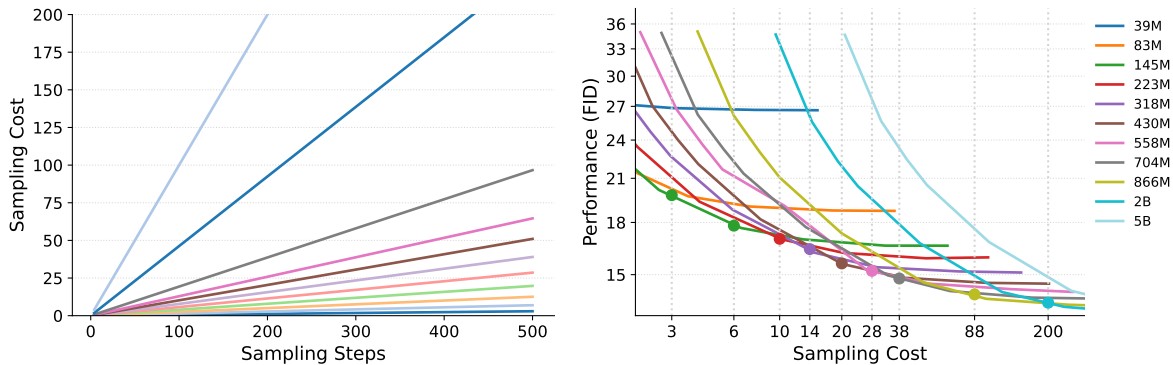

Figure 9: Comparison of text-to-image performance of models with varying sizes. The left figure shows the relationship between sampling cost (normalized cost × sampling steps) and sampling steps for different model sizes. The right figure plots the optimal text-to-image FID score among CFG rates of $(1.5, 2.0, 3.0, 4.0, 5.0, 6.0, 7.0, 8.0)$ as a function of the sampling cost for the same models. Key Observation: Smaller models achieve better FID scores than larger models for a fixed sampling cost. For instance, at a cost of 3, the 83M model achieves the best FID compared to the larger models. This suggests that smaller models can be more efficient in achieving good results with lower costs.

In this study, we find that CFG rate as a sampling parameter yields inconsistent results across different model sizes. Hence, it is interesting to quantitatively determine the *optimal* CFG rate for each model size and sampling steps using either FID or CLIP score. We demonstrate this by sampling the scaled models using different CFG rates, *i.e.*, $(1.5, 2.0, 3.0, 4.0, 5.0, 6.0, 7.0, 8.0)$ and comparing their quantitative and qualitative results. In Fig. 7, we present visual results of two models under varying CFG rates, highlighting the impact on the visual quality. We observed that changes in CFG rates impact visual quality more significantly than prompt semantic accuracy and therefore opted to use the FID score for quantitative determination of the optimal CFG rate. performance. Fig. 8 shows how different classifier-free guidance rates affect the FID scores in text-to-image generation (see figure caption for more details).

### 3.3.2 Scaling efficiency trends.

Using the optimal CFG rates established for each model at various number of sampling steps, we analyze the optimal performance to understand the sampling efficiency of different LDM sizes. Specifically, in Fig. 9, we present a comparison between different models and their optimal performance given the sampling cost

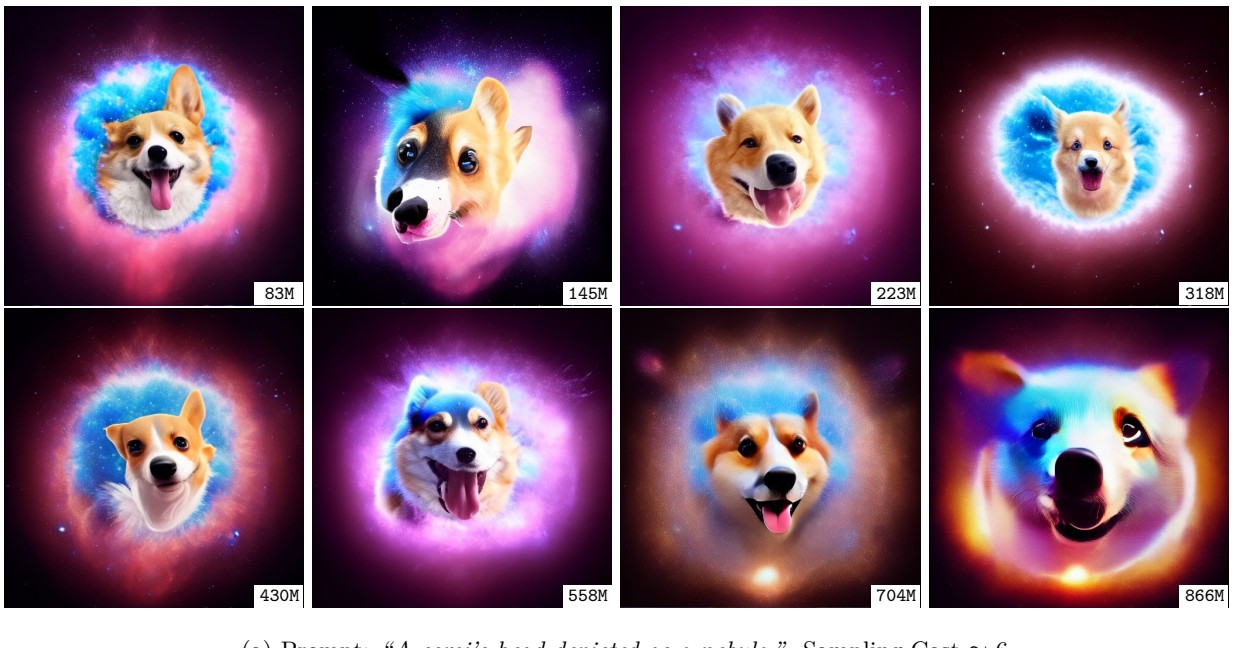

(a) Prompt: *"A corgi's head depicted as a nebula."*. Sampling Cost ≈ 6.

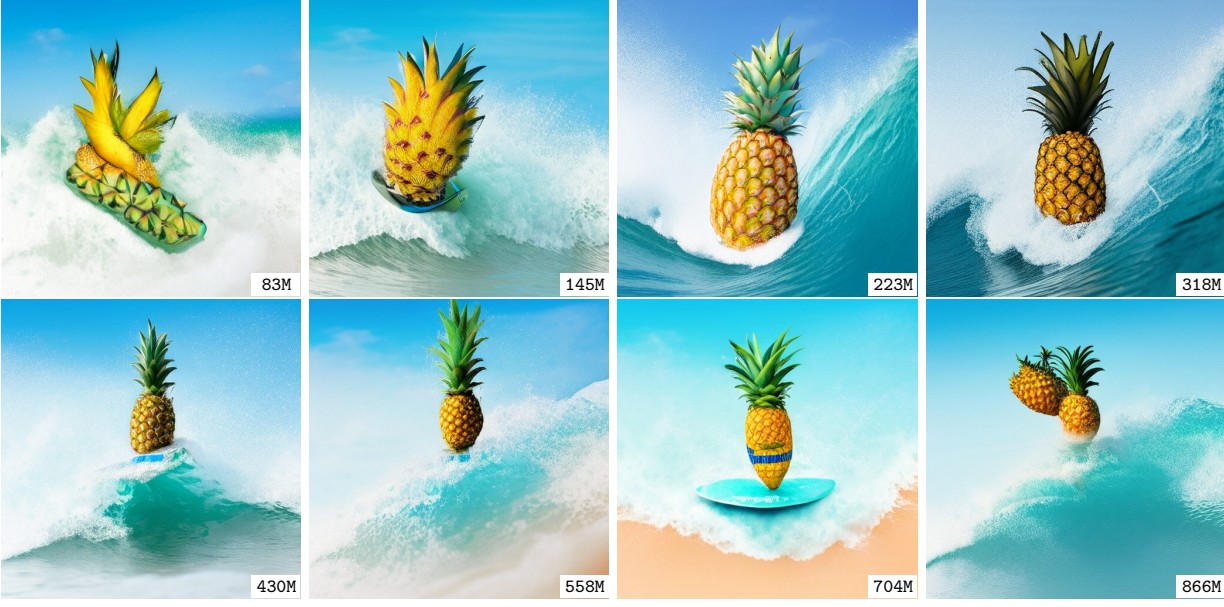

(b) Prompt: *"A pineapple surfing on a wave."*. Sampling Cost ≈ 12.

Figure 10: Text-to-image results of the scaled LDMs under approximately the same inference cost (normalized cost × sampling steps). Smaller models can produce comparable or even better visual results than larger models under similar sampling cost.

(normalized cost × sampling steps). By tracing the points of optimal performance across various sampling cost—represented by the dashed vertical line—we observe a consistent trend: smaller models frequently outperform larger models across a range of sampling cost in terms of FID scores. Furthermore, to visually substantiate better-quality results generated by smaller models against larger ones, Fig. 10 compares the results of different scaled models, which highlights that the performance of smaller models can indeed match their larger counterparts under similar sampling cost conditions. *Please see our supplement for more visual comparisons.*

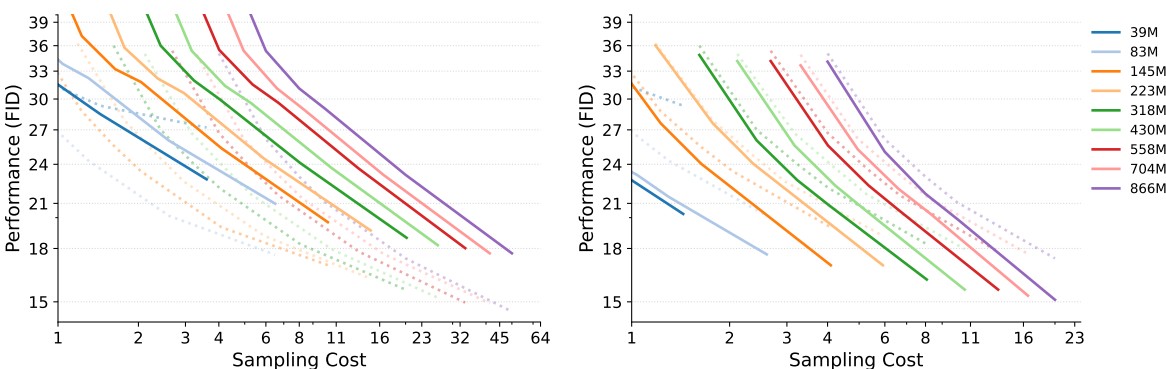

Figure 11: *Left*: Text-to-image performance FID as a function of the sampling cost (normalized cost × sampling steps) for the DDPM sampler (solid curves) and the DDIM sampler (dashed curves). *Right*: Text-to-image performance FID as a function of the sampling cost for the second-order DPM-Solver++ sampler (solid curves) and the DDIM sampler (dashed curves). Suggested by the trends shown in Fig. 9, we only show the sampling steps ≤ 50 as using more steps does not improve the performance.

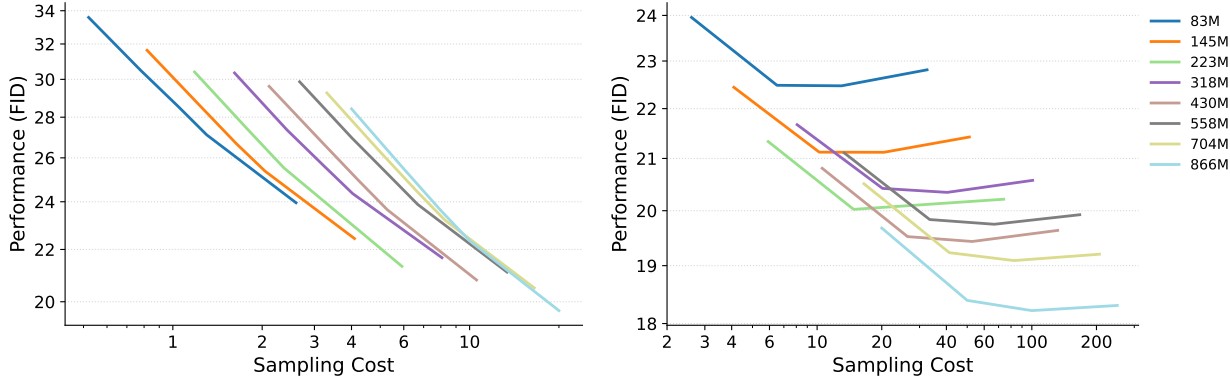

Figure 12: Super-resolution performance vs. sampling cost for different model sizes. *Left:* FID scores of super-resolution models under limited sampling steps (less than or equal to 20). Smaller models tend to achieve lower (better) FID scores within this range. *Right:* FID scores of super-resolution models under a larger number of sampling steps (greater than 20). Performance differences between models become less pronounced as sampling steps increase.

### 3.4 Scaling sampling-efficiency in different samplers

To assess the generalizability of observed scaling trends in sampling efficiency, we compared scaled LDM performance using different diffusion samplers. In addition to the default DDIM sampler, we employed two representative alternatives: the stochastic DDPM sampler (Ho et al., 2020) and the high-order DPM-Solver++ (Lu et al., 2022b).

Experiments illustrated in Fig. 11 reveal that the DDPM sampler typically produces lower-quality results than DDIM with fewer sampling steps, while the DPM-Solver++ sampler generally outperforms DDIM in image quality (see the figure caption for details). Importantly, we observe consistent sampling-efficiency trends with the DDPM and DPM-Solver++ sampler as seen with the default DDIM: smaller models tend to achieve better performance than larger models under the same sampling cost. Since the DPM-Solver++ sampler is not designed for use beyond 20 steps, we focused our testing within this range. This finding demonstrates that the scaling properties of LDMs remain consistent regardless of the diffusion sampler used.

### 3.5 Scaling downstream sampling-efficiency

Here, we investigate the scaling sampling-efficiency of LDMs on downstream tasks, specifically focusing on the super-resolution task. Unlike our earlier discussions on optimal sampling performance, there is limited

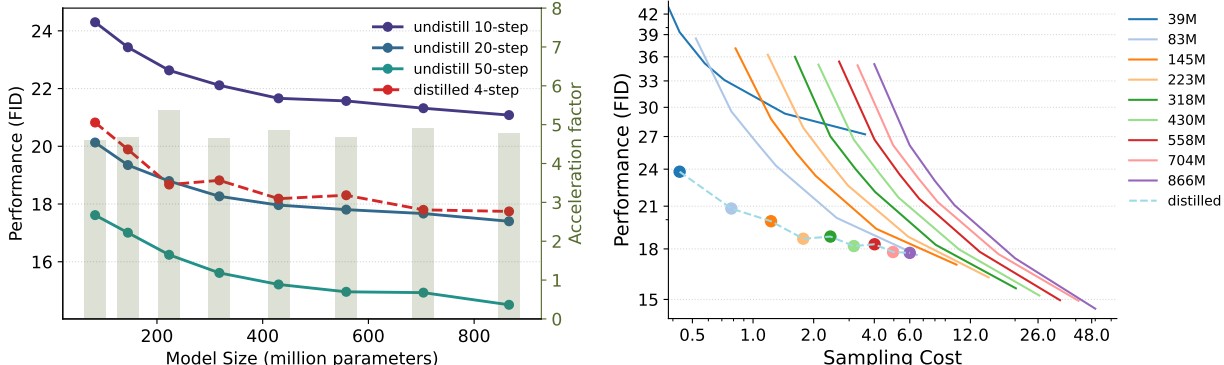

Figure 13: Distillation improves text-to-image performance and scalability. *Left:* Distilled Latent Diffusion Models (LDMs) consistently exhibit lower (better) FID scores compared to their undistilled counterparts across varying model sizes. The consistent acceleration factor (approx. 5×) indicates that the benefits of distillation scale well with model size. *Right:* Distilled models using only 4 sampling steps achieve FID scores comparable to undistilled models using significantly more steps. Interestingly, at a sampling cost of 7, the distilled `866M` model performs similarly to the smaller, undistilled `83M` model, suggesting improved efficiency.

literature demonstrating the positive impacts of SR performance without using classifier-free guidance. Thus, our approach directly uses the SR sampling result without applying classifier-free guidance. Inspired from Fig. 4, where the scaled downstream LDMs have significant performance difference in 50-step sampling, we investigate sampling efficiency from two different aspects, *i.e.*, fewer sampling steps [4, 20] and more sampling steps (20, 250]. As shown in the left part of Fig. 12, the scaling sampling-efficiency still holds in the SR tasks when the number of sampling steps is less than or equal to 20 steps. Beyond this threshold, however, larger models demonstrate greater sampling-efficiency than smaller models, as illustrated in the right part of Fig. 12. This observation suggests the consistent sampling efficiency of scaled models on fewer sampling steps from text-to-image generation to super-resolution tasks.

## 3.6 Scaling sampling-efficiency in distilled LDMs.

We have featured the scaling sampling-efficiency of latent diffusion models, which demonstrates that smaller model sizes exhibit higher sampling efficiency. A notable caveat, however, is that smaller models typically imply reduced modeling capability. This poses a challenge for recent diffusion distillation methods (Luhman & Luhman, 2021; Salimans & Ho, 2022; Song et al., 2023; Sauer et al., 2023b; Gu et al., 2023; Mei et al., 2024a; Luo et al., 2023; Lin et al., 2024) that heavily depend on modeling capability. One might expect a contradictory conclusion and believe the distilled large models sample faster than distilled small models. In order to demonstrate the sampling efficiency of scaled models after distillation, we distill our previously scaled models with conditional consistency distillation (Song et al., 2023; Mei et al., 2024a) on text-to-image data and compare those distilled models on their optimal performance.

To elaborate, we test all distilled models with the same 4-step sampling, which is shown to be able to achieve the best sampling performance; we then compare each distilled model with the undistilled one on the normalized sampling cost. We follow the same practice discussed in Section 3.3.1 for selecting the optimal CFG rate and compare them under the same relative inference cost. The results shown in the left part of Fig. 13 demonstrate that distillation significantly improves the generative performance for all models in 4-step sampling, with FID improvements across the board. By comparing these distilled models with the undistilled models in the right part of Fig. 13, we demonstrate that distilled models outperform undistilled models at the same sampling cost. However, at the specific sampling cost, *i.e.*, sampling cost ≈ 8, the smaller undistilled 83M model still achieves similar performance to the larger distilled 866M model. The observation further supports our proposed scaling sampling-efficiency after diffusion distillation.

## 4    Conclusion

In this paper, we investigated scaling properties of Latent Diffusion Models (LDMs), specifically through scaling model size from 39 million to 5 billion parameters. We trained these scaled models from scratch on a web-scale text-to-image dataset and then finetuned the pretrained models for downstream tasks. Our findings unveil that, under identical sampling costs, smaller models frequently outperform larger models, suggesting a promising direction for accelerating LDMs in terms of model size. We further show that the sampling efficiency is consistent in multiple axes. For example, it is invariant to various diffusion samplers (stochastic and deterministic), and also holds true for distilled models. We believe this analysis of scaling sampling efficiency would be instrumental in guiding future developments of LDMs, specifically for balancing model size against performance and efficiency in a broad spectrum of practical applications.

**Limitations and future work.**    This work utilizes visual quality inspection alongside established metrics like FID and CLIP scores. We opted to avoid human evaluations due to the immense number of different combinations needed for the more than 1000 variants considered in this study. However, it is important to acknowledge the potential discrepancy between visual quality and quantitative metrics, which is actively discussed in recent works (Zhang et al., 2021; Jayasumana et al., 2024; Cho et al., 2023).

Claims regarding the scalability of latent diffusion models are made specifically for the particular model family studied in this work (Rombach et al., 2022). Extending this analysis to other model families, particularly those incorporating transformer-based backbones such as DiT (Peebles & Xie, 2023; Mei et al., 2023), SiT (Ma et al., 2024), MM-DiT (Esser et al., 2024), and DiS (Fei et al., 2024), and cascaded diffusion models such as Imagen3 (Baldridge et al., 2024) and Stable Cascade (Pernias et al., 2023), would be a valuable direction for future research.

## 5    Acknowledgments

Vishal M. Patel was supported by NSF CAREER award 2045489. We are grateful to Keren Ye, Jason Baldridge, Kelvin Chan for their valuable feedback. We also extend our gratitude to Shlomi Fruchter, Kevin Murphy, Mohammad Babaeizadeh, and Han Zhang for their instrumental contributions in facilitating the initial implementation of the latent diffusion models.

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
