## A Scaling the text-to-image performance

In order to provide detailed visual comparisons for Fig. 1 in the main manuscript, Fig. 14, Fig. 15, and Fig. 16 show the generated results with the same prompt and the same sampling parameters (*i.e.*, 50-step DDIM sampling and 7.5 CFG rate).

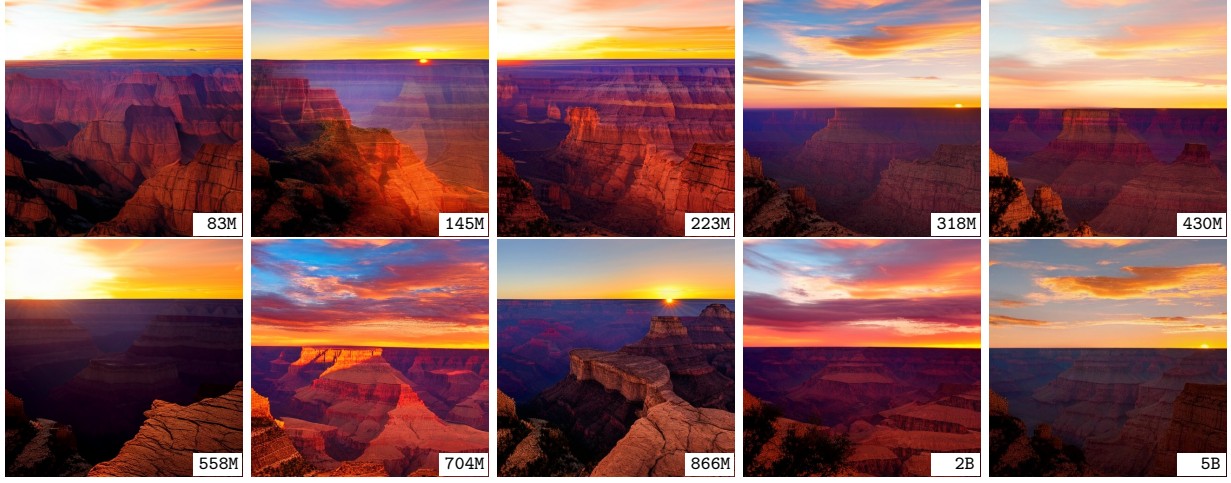

(a) Prompt: *"a professional photo of a sunset behind the grand canyon."*

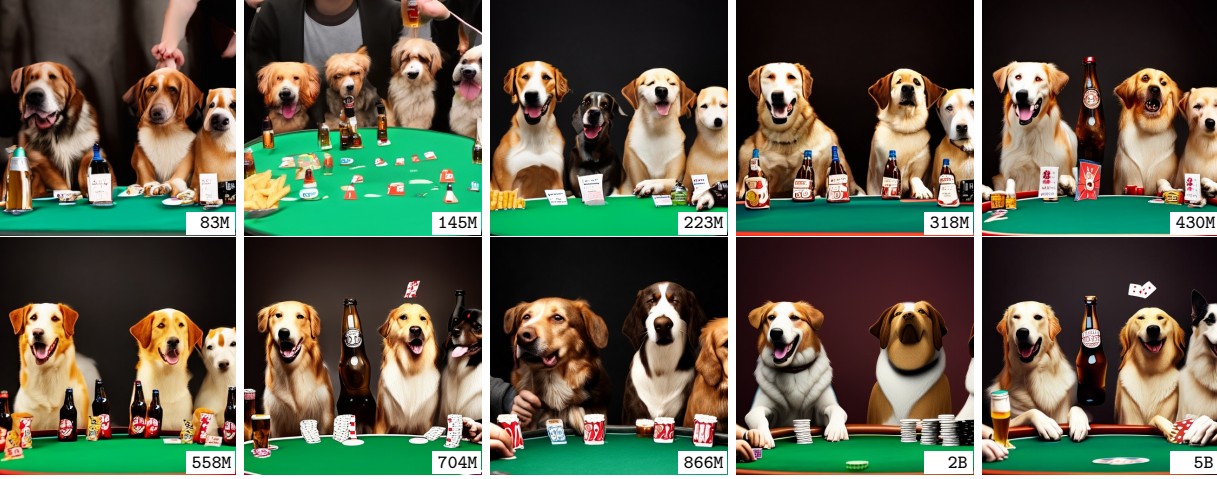

(b) Prompt: *"Dogs sitting around a poker table with beer bottles and chips. Their hands are holding cards."*

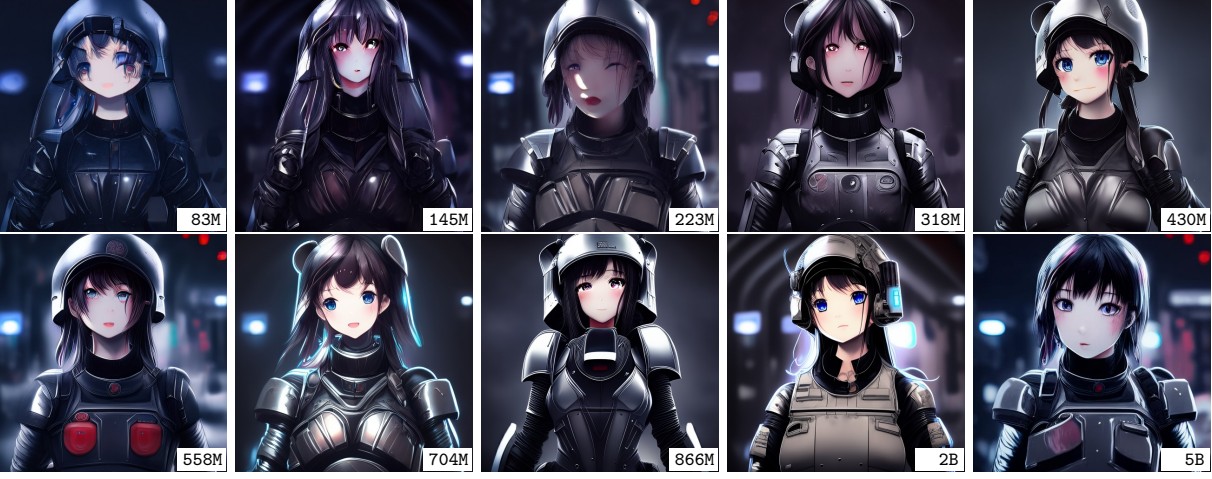

(c) Prompt: *'Portrait of anime girl in mechanic armor in night Tokyo."*

Figure 14: Text-to-image results from our scaled LDMs (83M - 5B), highlighting the improvement in visual quality with increased model size.

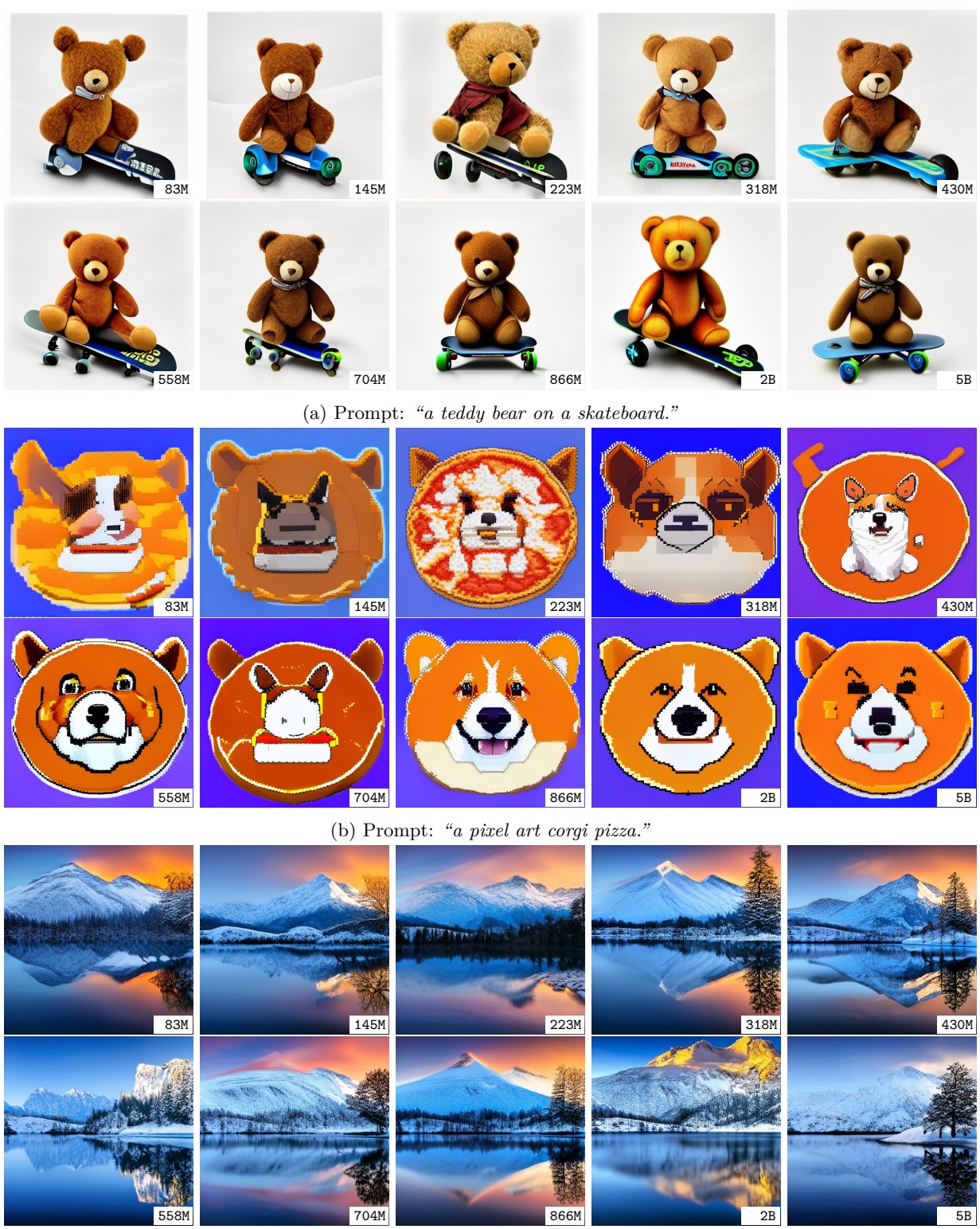

(a) Prompt: *"a teddy bear on a skateboard."*

(b) Prompt: *"a pixel art corgi pizza."*

(c) Prompt: *"Snow mountain and tree reflection in the lake."*

Figure 15: Text-to-image results from our scaled LDMs (83M - 5B), highlighting the improvement in visual quality with increased model size.

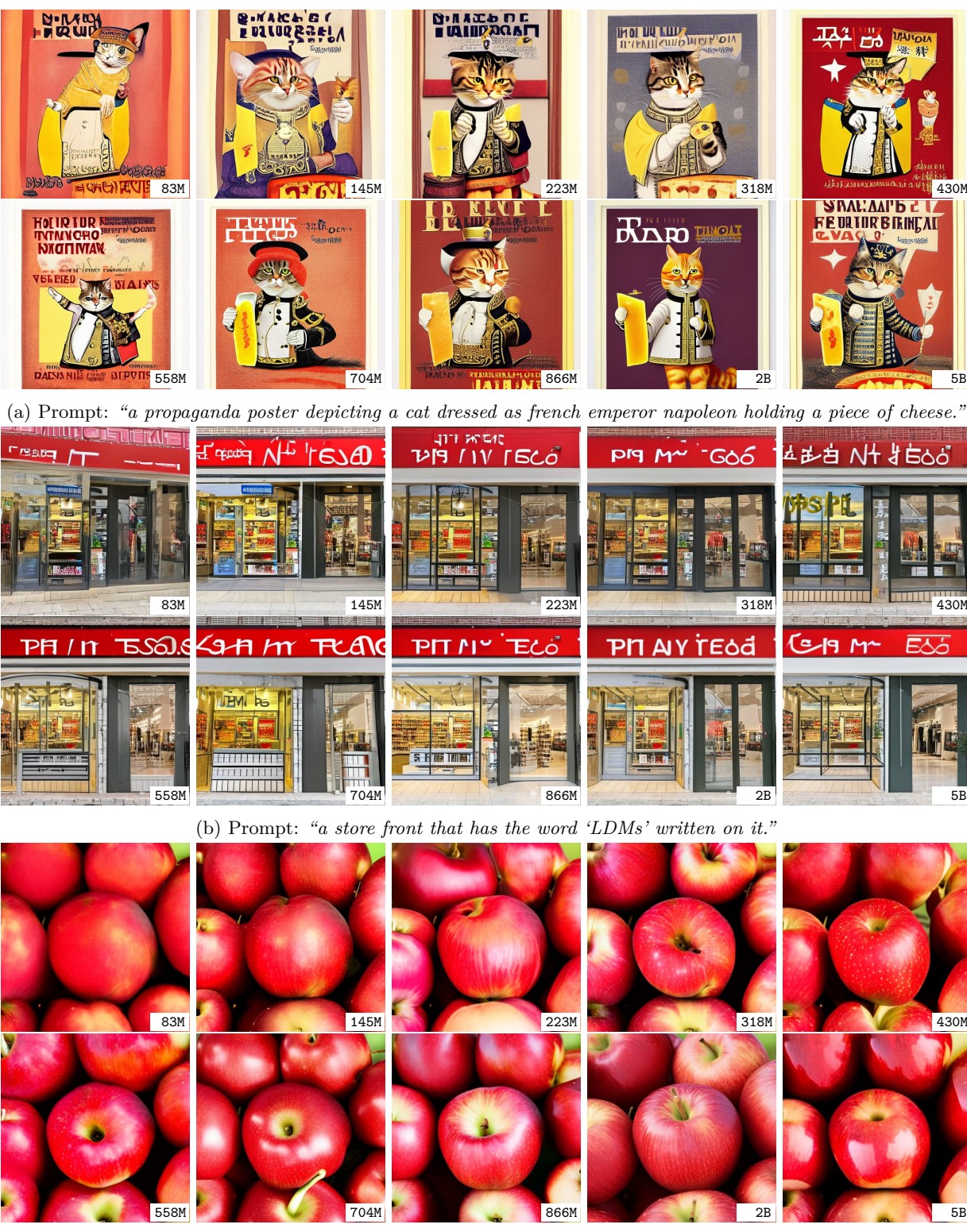

(a) Prompt: *"a propaganda poster depicting a cat dressed as french emperor napoleon holding a piece of cheese."*

(b) Prompt: *"a store front that has the word 'LDMs' written on it."*

(c) Prompt: *"ten red apples."*

Figure 16: Text-to-image results from our scaled LDMs (83M - 5B), highlighting the improvement in visual quality with increased model size.

# B  Scaling downstream performance

To provide more metrics for the super-resolution experiments in Fig. 4 of the main manuscript, Fig. 18 shows the generative metric IS for the super-resolution results. Fig. 18 shows the visual results of the super-resolution results in order to provide more visual results for the visual comparisons of Fig. 5 in the main manuscript.

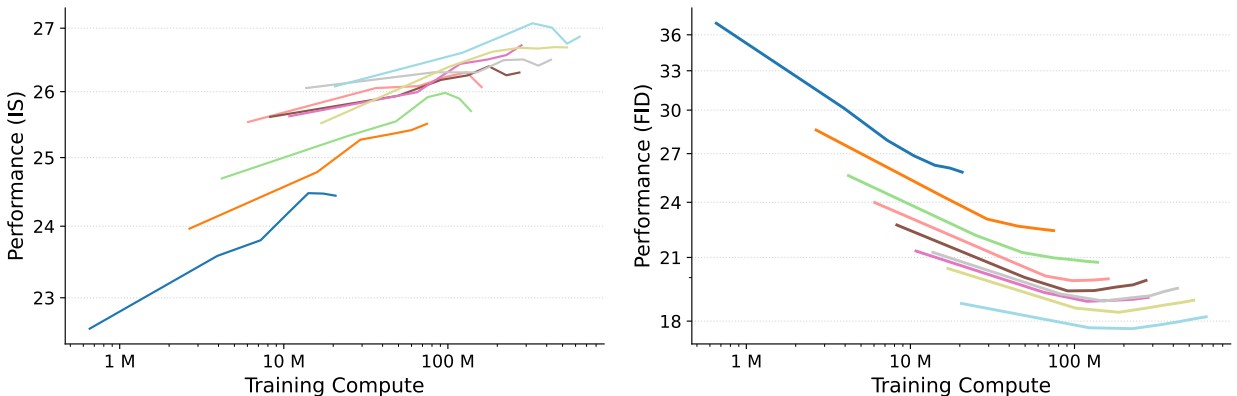

Figure 17: For super-resolution, we show the trends between the generative metric IS and the training compute still depend on the pretraining, which is similar to the trends between the generative metric FID and the training compute.

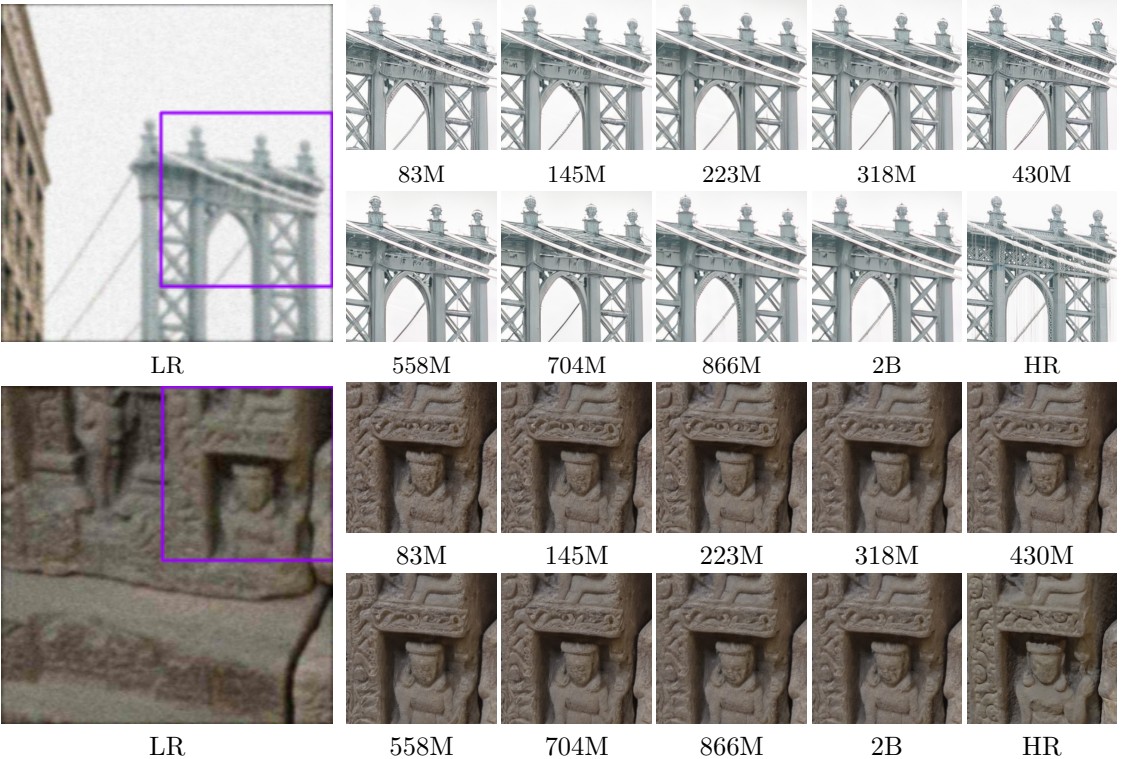

Figure 18: In 4× super-resolution, visual quality directly improves with increased model size. As these scaled models vary in pretraining performance, the results clearly demonstrate that pretraining boosts super-resolution capabilities.

## C   Scaling sampling-efficiency in distilled LDMs

Diffusion distillation methods for accelerating sampling are generally derived from Progressive Distillation (PD) Salimans & Ho (2022) and Consistency Models (CM) Song et al. (2023). In the main paper, we have shown that CoDi Mei et al. (2024a) based on CM is scalable to different model sizes. Here we show other investigated methods, *i.e.*, guided distillation Meng et al. (2023), has inconsistent acceleration effects across different model sizes. Fig. 19 shows guided distillation results for the 83M and 223M models respectively, where `s16` and `s8` denote different distillation stages. It is easy to see that the performance improvement of these two models is inconsistent.

Fig. 20 shows the visual results of the CoDi distilled models and the undistilled models under the same sampling cost to demonstrate the sampling-efficiency.

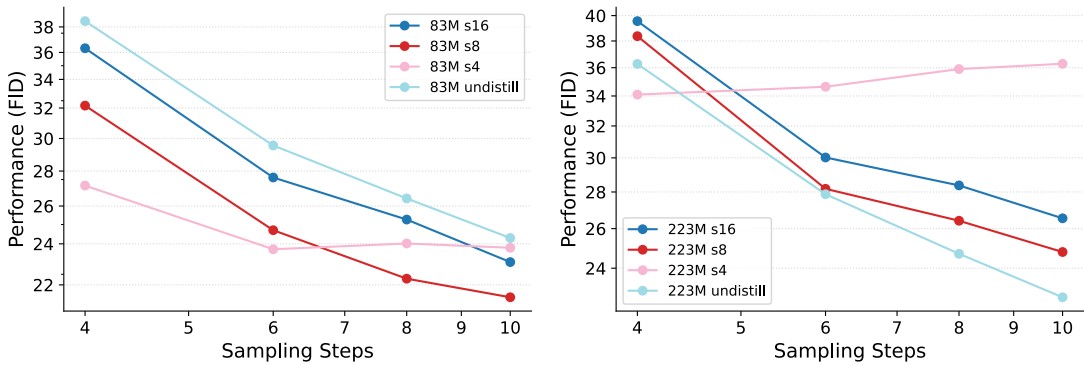

Figure 19: *Left:* Guided distillation on the 83M model for text-to-image generation. *Right:* Guided distillation on the 224M model for text-to-image generation.

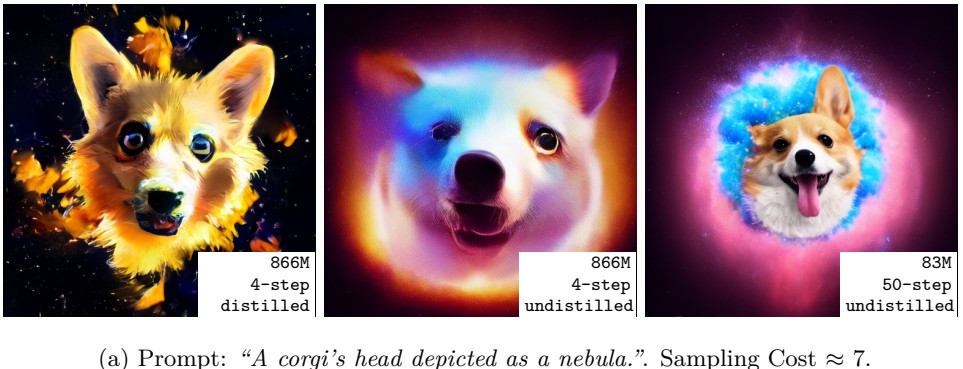

(a) Prompt: *"A corgi's head depicted as a nebula.".* Sampling Cost ≈ 7.

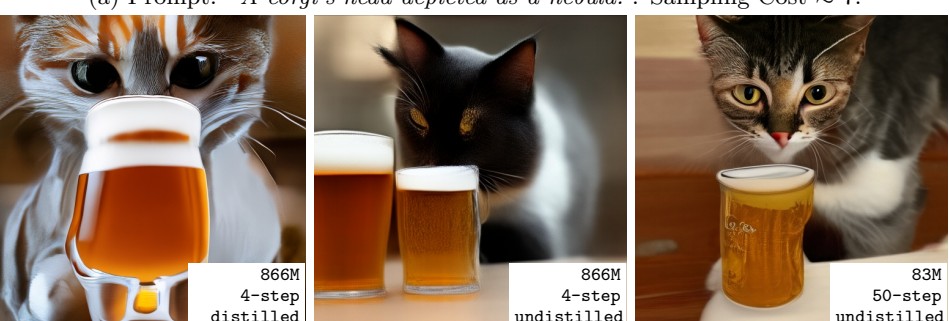

(b) Prompt: *"a cat drinking a pint of beer.".* Sampling Cost ≈ 7.

Figure 20: We visualize text-to-image generation results of the tested LDMs under approximately the same inference cost.

# D   Scaling the sampling-efficiency

To provide more visual comparisons additional to Fig. 10 in the main paper, Fig. 21, Fig.22, and Fig. 23 present visual comparisons between different scaled models under a uniform sampling cost. This highlights that the performance of smaller models can indeed match their larger counterparts under similar sampling cost.

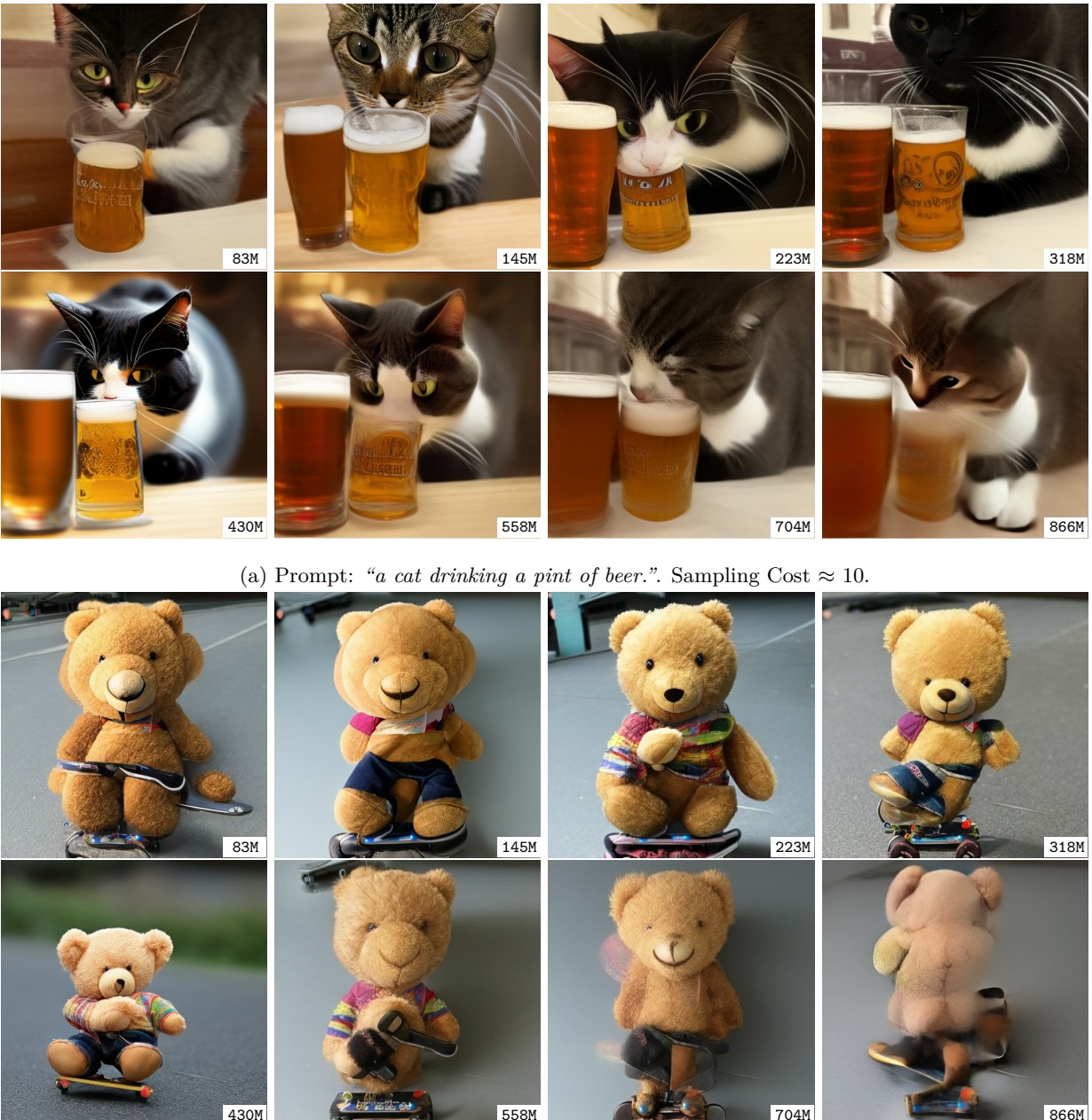

(a) Prompt: *"a cat drinking a pint of beer."*. Sampling Cost ≈ 10.

(b) Prompt: *"a teddy bear on a skateboard."*. Sampling Cost ≈ 10.

Figure 21: We visualize text-to-image generation results of the tested LDMs under approximately the same inference cost. We observe that smaller models can produce comparable or even better visual results than larger models under similar sampling cost (model GFLOPs × sampling steps).

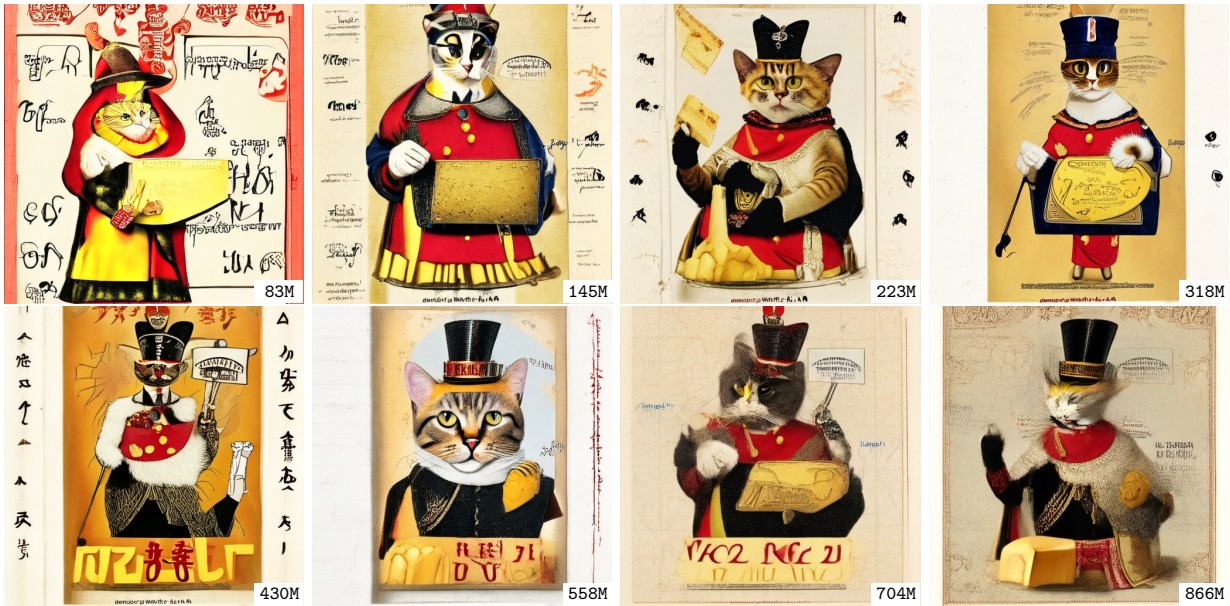

(a) Prompt: *"a propaganda poster depicting a cat dressed as french emperor napoleon holding a piece of cheese.".* Sampling Cost ≈ 14.

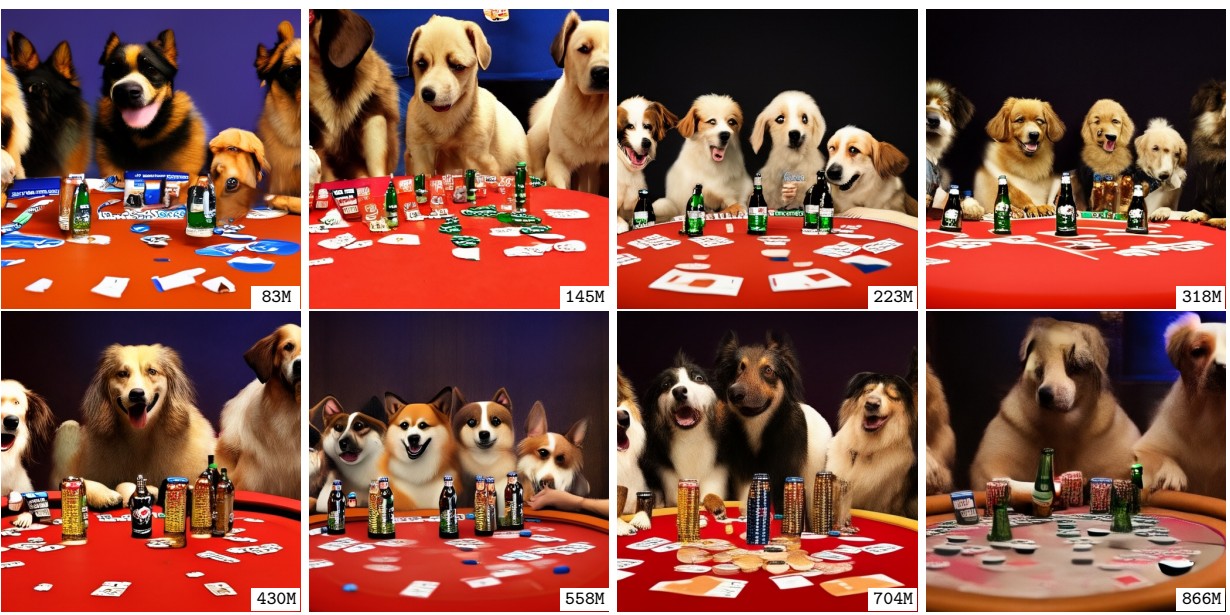

(b) Prompt: *"Dogs sitting around a poker table with beer bottles and chips.".* Sampling Cost ≈ 14.

Figure 22: We visualize text-to-image generation results of the tested LDMs under approximately the same inference cost. We observe that smaller models can produce comparable or even better visual results than larger models under similar sampling cost (model GFLOPs × sampling steps).

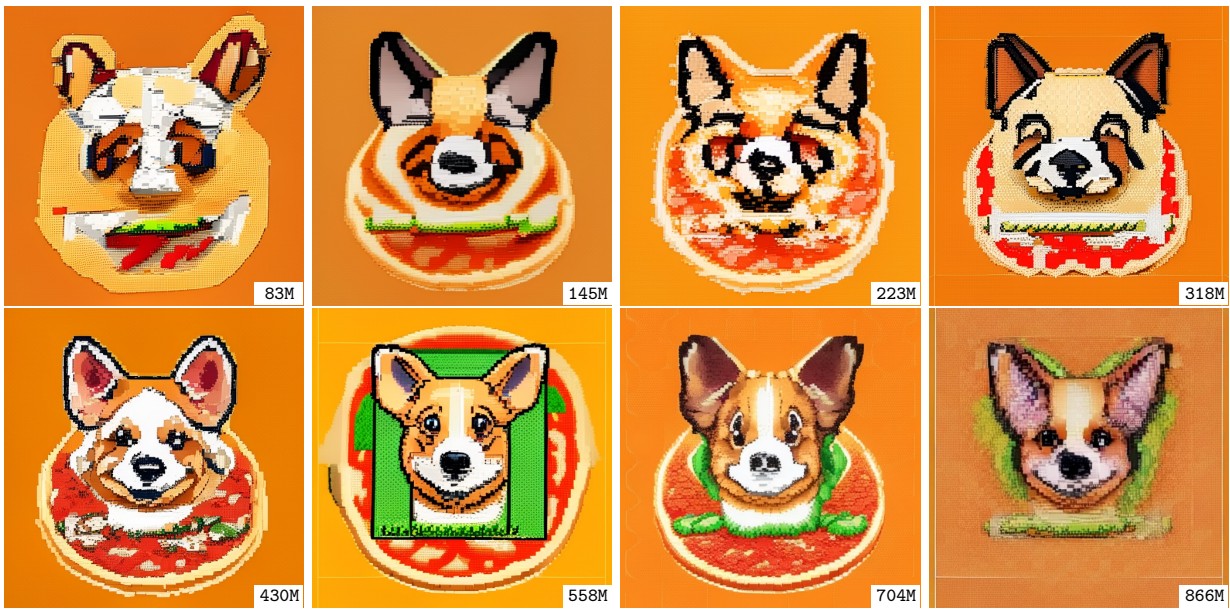

(a) Prompt: *"a pixel art corgi pizza."*. Sampling Cost ≈ 18.

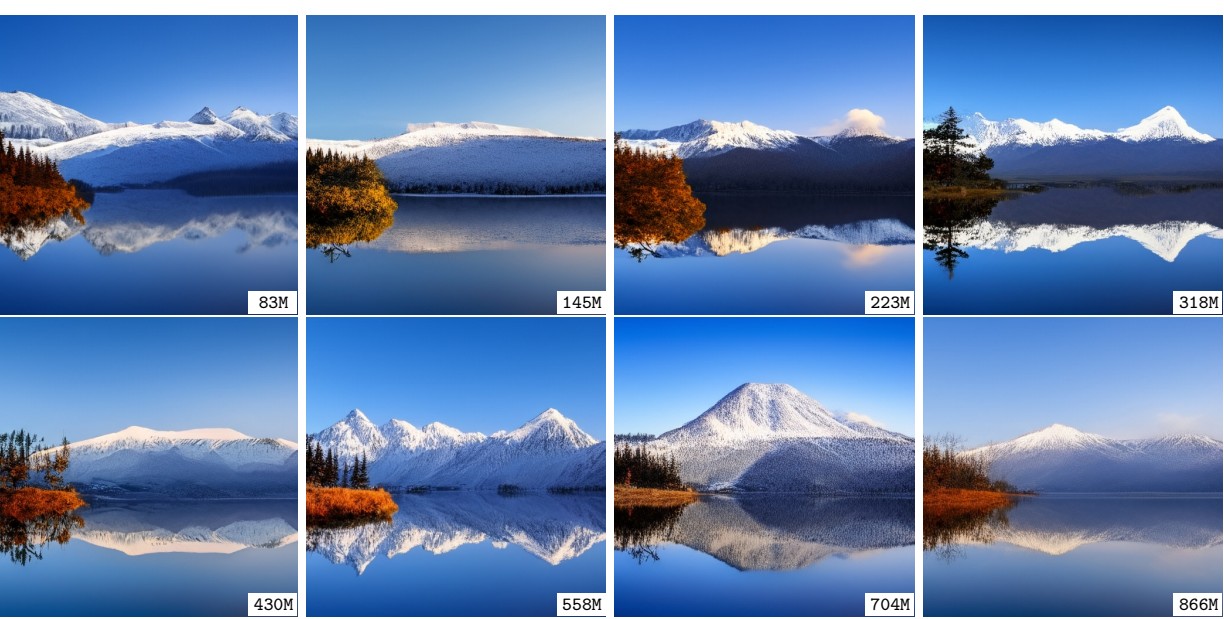

(b) Prompt: *"Snow mountain and tree reflection in the lake."*. Sampling Cost ≈ 18.

Figure 23: We visualize text-to-image generation results of the tested LDMs under approximately the same inference cost. We observe that smaller models can produce comparable or even better visual results than larger models under similar sampling cost (model GFLOPs × sampling steps).

# E    Scaling interpretability of text prompt interpolatation

Text prompt interpolation is widely recognized as a way to evaluate the interpretability of text-to-image models in recent works (Li et al., 2024; Park et al., 2023). In Figure 24, we show the text-prompt interopolation results of models in different sizes and visualize their sampling results. Specifically, we use two distinct prompts $A$ and $B$ and interpolate their CLIP embeddings as $\alpha A + (1 - \alpha)B, \alpha \in [0, 1]$, to generate intermediate text-to-image results. A clear pattern emerges: larger models leads to more semantically coherent and visually plausible interpolations compared to their smaller counterparts. The figure demonstrates the 2B model's superior ability to accurately interpret interpolated prompts, as evidenced by its generation of a tablet computer with a touch pen.

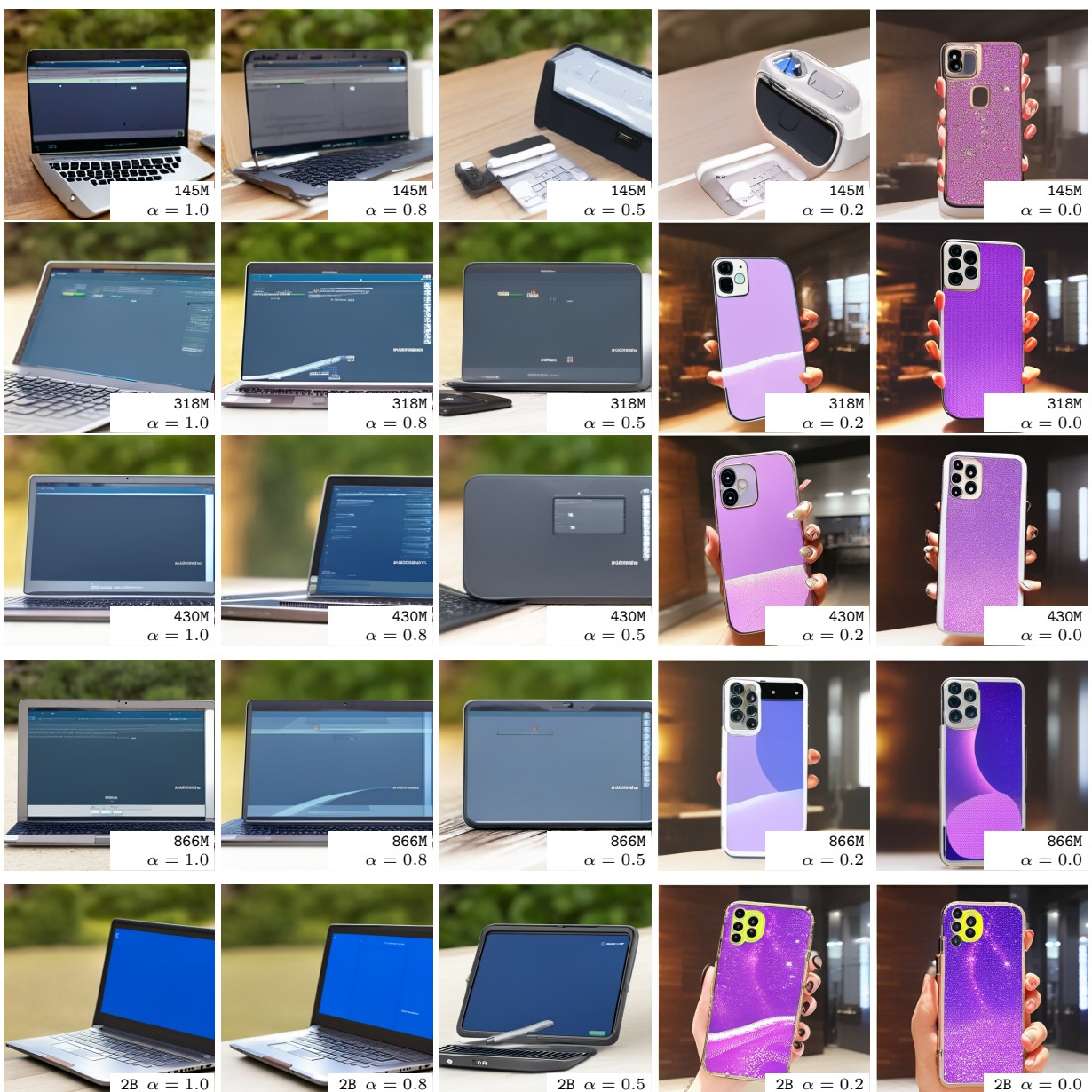

(a) Prompt $A$: *"a old computer"*. Prompt $B$: *"a fancy mobile phone"*.

Figure 24: We visualize the text-prompt interpolation results of scaled models in different sizes. Each row shows the results of the same model with different interpolation fraction $\alpha A + (1 - \alpha)B$. All results are sampled with the same 20-step DDIM sampler and CFG of 7.5.