# OpenReview forum: "Bigger is not Always Better: Scaling Properties of Latent Diffusion Models"
_TMLR — Accepted by TMLR_

### Review · Reviewer_z3RN · 2024-08-08

**Summary Of Contributions:**

This paper provides a thorough empirical analysis of the scaling properties of text-to-image diffusion models. Surprisingly, it is found that when operating under a given inference budget, smaller models frequently outperform their larger equivalents in generating high-quality results. Furthermore, the experiment results also show that distilled models outperform undistilled models at the same sampling cost. The resulting scaling properties of sampling efficiency could be a useful guideline of developing LDMs.

**Audience:**

Yes

**Broader Impact Concerns:**

No ethical concerns for this paper.

**Claims And Evidence:**

Yes

**Requested Changes:**

No significant change is requested, just some mild suggestions:
It would be much better and beneficial to the readers if the paper could also provide some theoretical analysis.
I'm also curious about if it could be extended to other model compressions methods.

**Strengths And Weaknesses:**

Strengths:
1. This paper provides a thorough empirical analysis of the scaling properties of text-to-image diffusion models.
2. Surprisingly, it is found that when operating under a given inference budget, smaller models frequently outperform their larger equivalents in generating high-quality results.
3. Furthermore, the experiment results also show that distilled models outperform undistilled models at the same sampling cost.
4. The resulting scaling properties of sampling efficiency could be a useful guideline of developing LDMs.

Weaknesses:
1. This paper is pure empirical, which lacks theoretical analysis to help explain the empirical results.
2. I also wonder whether the conclusion will still hold if we replace distillation with other model compression methods such as quantization or sparsification.

---

> ### Author Response · Authors · 2024-09-26
> **Response to z3RN**
>
> We thank the reviewer for the valuable feedback. We revised the manuscript accordingly and highlighted the changes in red.
>
> **Theoretical analysis**: We acknowledge the reviewer's point regarding the lack of theoretical analysis. Our primary focus in this work was to establish a comprehensive empirical understanding of the scaling properties of LDMs. We believe that these empirical findings are valuable in their own right, providing a solid foundation for future theoretical exploration.
>
> **Model Quantization / Compression**: While model quantization can impact performance, the specific technique used has a greater effect than the choice of model.  This aspect, however, falls outside the scope of our current work, which prioritizes investigating the scalability of models. We believe the interplay between quantization and scalability represents a promising avenue for future research.

---

### Review · Reviewer_uzX7 · 2024-08-20

**Summary Of Contributions:**

This paper brings a fresh perspective on the common belief that large model size always leads to better performance. The authors provide empirical analyses of Latent Diffusion Models (LDM), focusing on how model scaling from small (39 million) to large (5 billion) parameter counts affect the sample efficiency and visual quality. The key findings from the empirical analyses can be summarized as

1. the scaled LDMs with various parameter sizes exhibit similar trends in generative performance relative to the training compute cost.

2. there is a strong correlation between the pretraining performance and success in downstream task.

3. small models can frequently outperform larger models across a range of sampling cost in terms of FID.

4. the observation in 3 holds under various settings: with different samplers (DDIM, DDPM and high-order DPM-Solver++), in the downstream super-resolution task and in distilled LDM.

**Audience:**

Yes

**Broader Impact Concerns:**

This paper primarily focuses on understanding the scaling properties of Latent Diffusion Models (LDMs) and how model size affects sampling efficiency and visual quality. The insights gained from this study could contribute to more efficient generative modeling techniques, particularly in scenarios where computational resources are limited. I don't see any negative ethical risk of the current work.

**Claims And Evidence:**

Yes

**Requested Changes:**

1. Including practical deployment scenarios where the benefits of smaller models are most evident would enhance the paper’s relevance to both academic and industry practitioners. These insights would make the research more directly actionable.

2. To make the analysis more comprehensive, the authors could consider evaluating other key performance metrics such as generalization ability, interpretability, and robustness. This would provide a more holistic view of the implications of model scaling.

**Strengths And Weaknesses:**

**Strengths:**

1. The key strength of the paper is its comprehensive exploration of model scalings in LDMs. It highlights that smaller models can have better performance and sample efficiency when the compute budget is constrained. It challenges the prevailing "bigger is better" mindset in deep learning and sheds insights on development of LDMs for balancing model size against performance and sample efficiency in the practical scenario when the compute budget is constraint.

2. The consistency of the sampling efficiency trends under different settings adds credibility to the claims made in the paper.

**Weaknesses:**

1. The paper focuses on sampling efficiency and visual quality. It could benefit from a broader analysis of other performance metrics such as generalization/robustness in different tasks.

2. Although empirical results are well-explored, there is not many intuitive explanations of the empirical findings. For instance, when presenting the scaling sampling-efficiency with different samplers, some arguments that explain the consistent trend in sampling-efficiency would help to validate the claim in the paper can possibly be generalized to other types of diffusion models.

---

> ### Author Response · Authors · 2024-09-26
> **Response to uzX7**
>
> We thank the reviewer for the valuable feedback. We revised the manuscript accordingly and highlighted the changes in red.
>
> **Generalization (Robustness) and Interpretability of Scaled LDMs**: We investigated the generalization and robustness of scaled latent diffusion models (LDMs) for image generation. Our analysis focused on how well models of different sizes performed on image super-resolution and DreamBooth fine-tuning (Section 3.2). Larger models demonstrated superior generalization, as evidenced by the improved performance of their fine-tuned versions compared to smaller models under the same sampling steps. Furthermore, larger models exhibited greater robustness because they showed better performance than smaller models at the same training compute Figure 4).
>
> Evaluating interpretability in text-to-image models is complex, lacking standardized tests. To assess this, we employed a novel approach: gradually interpolating between two text prompts and observing the corresponding image transitions (Supplementary Section E). Our findings indicate that larger models generate smoother and more logical transitions, suggesting a higher degree of interpretability.
>
> **Benefit on Practical Deployment Scenarios**: Our findings on scaling properties have significant implications for real-world applications with limited computational resources, such as mobile and edge computing. We demonstrate that smaller models,, can achieve superior performance compared to larger models when given the same computational budget. This provides a practical guideline for model selection in resource-constrained environments

---

### Review · Reviewer_uGCq · 2024-09-15

**Summary Of Contributions:**

This paper investigates the scaling properties of latent diffusion models (LDMs) with a focus on sampling efficiency. Through empirical analysis, the study reveals that smaller models often outperform larger ones in generating high-quality results under a given inference budget. The findings are further validated across various diffusion samplers, downstream tasks, post-distilled models, and training compute comparisons. These insights suggest new strategies for optimizing LDMs to enhance generative capabilities within limited inference budgets.

**Audience:**

Yes

**Claims And Evidence:**

Yes

**Requested Changes:**

The experimental part need to be clarified based to the comments above.

**Strengths And Weaknesses:**

Pros:

- The paper presents extensive experiments to substantiate the claims made in the introduction.
- It provides insights into the scaling laws of visual generation, highlighting the differences between vision and language scaling laws.
- The author's conclusion is intriguing—it's fascinating to see that smaller models can yield good performance given a specific inference budget. This counter-intuitive finding is particularly interesting.

Cons:

- In Table 1, is the CFG value consistent across all experiments? This should be clearly stated in the caption.
- The scaling ablation in Figure 2 is conducted on U-Net, which is known to scale poorly in terms of parameter number and performance. Could the authors perform the same experiment using Stable Diffusion v3, given its pure transformer-based structure? This could also help explore flow matching-based samplers, as previously mentioned.
- As shown in the caption of Figure 8, are the results in Figures 11 and 12 obtained using the FID from the optimal CFG value derived from Figure 8? This needs to be explicitly stated.
- In the conclusion, it would be valuable to know if the same findings apply to cascaded diffusion models, not just LDMs. For future work on the backbone, it would be interesting to investigate whether the same conclusions hold for Mamba-based backbones, DiS, and ZigMa.

Dis: Scalable Diffusion Models with State Space Backbone

Zigma: ZigMa: A DiT-style Zigzag Mamba Diffusion Model,ECCV24

---

> ### Author Response · Authors · 2024-09-26
> **Response to uGCq**
>
> We thank the reviewer for the valuable feedback. We revised the manuscript accordingly and have highlighted the changes in red.
>
> **Revised Statement about CFG Rates in Table.1**: To ensure clarity and consistency, we have made the following revisions:
> - Table 1: The caption now explicitly states the CFG rates used, which are consistent across all models in the table.
> - Figure 1, 3, 4, 5, 6, and 7: The captions now specify both the sampling steps and CFG rates used in each figure.
>
> **Experiments on Stable Diffusion 3 (Rectified Flow Transformers)**: While we appreciate the suggestion to include experiments on Stable Diffusion 3 and its use of Rectified Flow Transformers (RFTs), this falls outside the scope of our current study. Our manuscript focuses specifically on investigating the scaling properties of Latent Diffusion Models (LDMs)  (Rombach et al., 2022), a representative family of generative models.  Fully replicating SD3 is currently infeasible due to the lack of publicly available training code. Therefore, we've opted for a focused approach to present the most robust evidence possible, aligning with TMLR's acceptance criteria.
>
> **CFG Rates in Figure 11 and 12**:
> Both Figure 11 and Figure 12 are irrelevant with Figure 8. We have updated their caption to clarify this.)
>
> - Figure 11: In Figure 11, for each model and sampling cost, we tested CFG rates from 1.0 to 8.0 and selected the rate that yielded the best (lowest) FID score with the DDPM sampler and DPM-Solver++. This figure demonstrates the robustness of our conclusion when using different diffusion samplers.
> - Figure 12: This figure examines FID values without using Classifier-Free Guidance (CFG). As discussed in Section 3.5, there's limited evidence that CFG consistently improves super-resolution (SR) performance. Therefore, we opted to analyze FID scores in this context without CFG.

---

### Author Response · Authors · 2024-09-26
**Summary of Changes**

We thank all reviewers for their valuable feedback and have addressed the major concerns raised. The changes in revision includes:
- CFG Rates: We clarified the respective CFG rates in all tables and figures.
- Cascaded Diffusion, DiS, ZigMa: We added discussions to other works.
- Interpretability: Introduced an interpretability analysis and presented results in Supplementary Section E.

---

### Decision · Action_Editor_ipts · 2024-11-17

**Recommendation:** Accept as is

**Comment:**

The reviewers unanimously recommend accepting this paper, with strong agreement on the quality and thoroughness of the empirical analysis of Latent Diffusion Models (LDMs). The work demonstrates that smaller models can outperform larger ones under fixed inference budgets, challenging the "bigger is better" paradigm. While the research is primarily empirical and focuses on U-Net backbones, the findings are well-substantiated across various settings including different samplers, downstream tasks, and distilled models. The authors have addressed the initial concerns.

The primary limitation noted is the strong focus on U-Net backbones, which may limit generalizability to newer architectures like Transformers, Mamba, and Linear-Attention models. In the author's response to reviewer uGCq who raised this concern, they acknowledged the suggestion to include experiments on Stable Diffusion 3 (which uses Transformers) but stated that this was "outside the scope of our current study" and that fully replicating SD3 was "infeasible due to the lack of publicly available training code." They chose to maintain their focus specifically on investigating the scaling properties of U-Net-based Latent Diffusion Models. While they added discussions about other architectures like DiS, ZigMa, and Cascaded Diffusion in their revision, they did not conduct empirical studies with these newer architectures. Therefore, this limitation remains a valid concern about the generalizability of their findings.

**Audience:**

TMLR's audience would be interested in these findings. The paper challenges a fundamental assumption in machine learning that larger models generally perform better, providing empirical evidence that smaller LDMs can outperform larger ones under fixed compute budgets. This has significant practical implications for deploying generative models in resource-constrained environments and could influence future research directions in model scaling and efficiency. While the work focuses on U-Net architectures, the underlying principles about compute-performance trade-offs are relevant to the broader machine learning community, particularly those working on generative models and model optimization.

**Claims And Evidence:**

The claims are well-supported by comprehensive empirical evidence. The authors conducted extensive experiments across different model sizes (from 39M to 5B parameters), various sampling methods (DDIM, DDPM, DPM-Solver++), and multiple downstream tasks. The results consistently demonstrate that smaller models can outperform larger ones under fixed compute budgets. The authors also strengthened their findings by addressing reviewer feedback with additional interpretability analysis, clear documentation of CFG rates, and robustness evaluations. The experimental methodology is transparent, and results are presented clearly with appropriate metrics like FID scores. While the study lacks theoretical analysis, the empirical evidence is thorough enough to support the paper's main claims about LDM scaling properties.